

# Neogene Caribbean elasmobranchs: Diversity, paleoecology and paleoenvironmental significance of the Cocinetas Basin assemblage (Guajira Peninsula, Colombia)

Jorge Domingo Carrillo–Briceño[1,2], Zoneibe Luz[3], Austin Hendy[4], László Kocsis[5], Orangel Aguilera[6], and Torsten Vennemann[3]

[1]Palaeontological Institute and Museum, University of Zurich, Karl–Schmid–Strasse 4, 8006 Zurich
[2]Smithsonian Tropical Research Institute, Av. Gorgas, Ed. 235, 0843–03092 Balboa, Ancón, Panamá
[3]Institut des Dynamiques de la Surface Terrestre, Université de Lausanne, Rue de la Mouline, 1015 Lausanne
[4]Natural History Museum of Los Angeles County, 900 Exposition Blvd, Los Angeles, California 90007 USA
[5]Universiti Brunei Darussalam, Faculty of Science, Geology Group, Jalan Tungku, BE 1410 Brunei Darussalam
[6]Departamento de Biologia Marinha, Instituto de Biologia, Universidade Federal Fluminense, Niterói, 24020–150 Rio de Janeiro, Brasil

**Correspondence:** Zoneibe Luz (zoneibe.luz@gmail.com)

**Abstract.** The Cocinetas Basin is located on the eastern flank of La Guajira Peninsula, northern Colombia (South Caribbean). During late Oligocene through Pliocene, much of the basin was submerged. The extensive deposits in this area suggest a transition from a shallow marine to a fluvio–deltaic system, with a rich record of invertebrate and vertebrate fauna. The elasmobranch assemblages of the early Miocene to late Pliocene succession in the Cocinetas Basin (Jimol, Castilletes and Ware Formations, and Patsúa Valley) are described for the first time. The assemblages include at least 30 taxa of sharks (Squaliformes, Pristio-
5  phoriformes, Orectolobiformes, Lamniformes and Carcharhiniformes) and batoids (Rhinopristiformes and Myliobatiformes), of which 24 taxa are reported from the Colombian Neogene for the first time. Paleoecological and paleoenvironmental interpretations are based on the feeding ecology, and on estimates of paleosalinity using stable isotope compositions of oxygen in the bioapatite of shark teeth. The isotopic composition of studied specimens corroborates the paleoenvironmental settings for the studied units suggested on the basis of other proxies. These Neogene elasmobranch assemblages from the Cocinetas
10  Basin, provide new insights of the shark and ray diversity inhabiting the coastal and estuarine environments of the northwestern margin of South America, both during the existence of the gateway between the Atlantic and Pacific Oceans, and following its closure.

## 1  Introduction

15  During the Neogene, large areas of the northern margin of South America were submerged (see Iturralde–Vinent and MacPhee, 1999) and influenced by the paleoceanographic connection between the Pacific and Atlantic oceans along the Central American Seaway (CAS). The CAS is defined here as a deep oceanic connection between the Pacific and Atlantic oceans along the tectonic boundary of Caribbean and the South American plates (Jaramillo et al., 2017). The CAS existed throughout the Cenozoic, but was reduced in width by the early Miocene (Farris et al., 2011), and the transfer of deep–water ceased by the




late Miocene 12–10 Ma (Montes et al., 2015; Bacon et al., 2015; Jaramillo et al., 2017). Shallow marine connections between Caribbean and Pacific waters existed until about 4.2–3.5 Ma, when a complete closure occurred (Coates and Stallard, 2013). The Cocinetas Basin, located on the eastern flank of La Guajira Peninsula, northern Colombia, records a transition in marine and terrestrial paleoenvironments during this regional change in conditions. This region presents extensive and well exposed

sedimentary deposits spanning the last 25 Myr (Moreno et al., 2015). The paleoenvironments are characterized by a transition from shallow marine deposits to a fluvio–deltaic system (Moreno et al., 2015), with a rich fossil record of invertebrates (Hendy et al., 2015) and vertebrates (Aguilera et al., 2013, 2017b; Moreno et al., 2015; Cadena and Jaramillo, 2015; Amson et al., 2016; Carrillo–Briceño et al., 2016b; Moreno–Bernal et al., 2016; Pérez et al., 2016). Ages for many of the fossiliferous units in the sequence have been estimated using Sr isotope stratigraphy (see Hendy et al., 2015).

Neogene marine chondrichthyan faunas from the southern Proto–Caribbean (especially from the northern margin of South America) are well known from Venezuela and some Lesser Antilles (e.g., Leriche, 1938, Casier, 1958, Casier, 1966, Aguilera, 2010, Aguilera and Lundberg, 2010, Carrillo–Briceño et al., 2015b, Carrillo–Briceño et al., 2016a, and references therein). But reports on chondrichthyans from the Neogene of Colombia are scarce. Previous reports from the Cocinetas Basin include fossil elasmobranchs without taxonomic description (Lockwood, 1965), a checklist of 14 families (Moreno et al., 2015), and

the description of a small assemblage of 13 taxa from the early Miocene Uitpa Formation (Carrillo–Briceño et al., 2016b).

A taxonomic revision is presented of the elasmobranch fauna collected in the Cocinetas Basin (Figs. 1–2), from the Jimol (Burdigalian), Castilletes (late Burdigalian–Langhian), Ware (Gelasian–Piacenzian) Formations, and two localities of the Patsúa Valley (Burdigalian–Langhian). The assemblage includes 30 taxa, of which 24 are new reports for Colombian Neogene deposits. Additionally, paleoecological and paleoenvironmental interpretations based on the feeding ecology of extant coun-

terpart species, as well as estimates of the paleosalinity using stable isotope compositions of oxygen in the bioapatite of shark teeth are discussed. The Cocinetas Basin represents a valuable window into dynamic changes in paleodiversity experienced by ancient Proto–Caribbean Neogene chondrichthyan faunas.

## 2 Material and Methods

The fossil elasmobranch assemblages (Table 1, Tables S1–S3; File S4) consists of 2529 specimens from 36 localities (Table S1)

from the Cocinetas Basin, Guajira Peninsula, northeastern Colombia (Fig. 1). The elasmobranch faunas were collected in the early Miocene Jimol Formation (six localities and 113 specimens), early–middle Miocene Castilletes Formation (20 localities and 1232 specimens), and the late Pliocene Ware Formation (eight localities and 215 specimens) (Tables S1–S2). Localities STRI 290468 and 290472 (968 specimens) in the Patsúa Valley, close to Flor de Guajira, along the southern margin of the Cocinetas Basin (Fig. 1), are from strata that cannot be readily correlated with either the Jimol or Castilletes Formation. Because

of these difficulties, and differences in their facies, and invertebrate and vertebrate fauna, we treat them as undifferentiated Jimol/Castilletes Formation, and they are referred to herein as the Patsúa assemblage.

The samples were collected by JDCB, AH and other collaborators during several expeditions between 2010 and 2014. Large specimens were collected directly from the outcrop, while 50 kg bulk sediment for subsequent picking of smaller specimens



were collected from the locality 290468 (Patsúa assemblage) and Castilletes Formation (localities 290632 and 390094). The bulk sediments were sieved and screen washed (mesh sizes: 0.5 and 2 mm). The overall Cocinetas Basin elasmobranch specimens (File S4) are housed in the paleontological collections of the Mapuka Museum of Universidad del Norte (MUN), Barranquilla, Colombia. The nomenclature follows Cappetta (2012) and Compagno (2005), with the exception of Rhinopristiformes

Last et al., 2016, Aetobatidae Agassiz, 1958 (Table 1) and *Carcharocles* Agassiz, 1838, for which we follow the nomenclature discussed in Last et al. (2016), White and Naylor (2016) and Ward and Bonavia (2001), respectively. Identifications are based on literature review (e.g., Santos and Travassos, 1960, Müller, 1999, Purdy et al., 2001, Cappetta, 1970, Cappetta, 2012, Reinecke et al., 2011, Reinecke et al., 2014, Voigt and Weber, 2011, Bor et al., 2012, Carrillo–Briceño et al., 2014, Carrillo–Briceño et al., 2015a, Carrillo–Briceño et al., 2015b, Carrillo–Briceño et al., 2016a, Aguilera et al., 2017a, among others) and comparative

analysis between fossil and extant specimens from several collections including Museu Paraense Emilio Goeldi (MPEG–V), Belém, Brazil; Fossil Vertebrate Section of the Museum für Naturkunde, Berlin, Germany (MB.Ma.); Natural History Museum of Basel (NMB), Switzerland; the paleontological collections of the Alcaldía del Municipio Urumaco (AMU–CURS) and Centro de Investigaciones Antropológicas, Arqueológicas y Paleontológicas of the Universidad Experimental Francisco de Miranda (CIAAP, UNEFM–PF), both in Venezuela; Paleontological collection of the Institut des Sciences de l'Evolution,

University of Montpellier (UM), France; Palaeontological Institute and Museum at the University of Zurich (PIMUZ), and René Kindlimann private collection, Uster, Switzerland.

Quantitative data includes percentages of specimens by order, families and genera recorded in the overall assemblages of the Cocinetas Basin (Table 1, Tables S1–S2, Fig. S5). Paleoecological interpretations of fossil chondrichthyan assemblages have limitations related to the scarce information offered by the fossil record. Extant sharks and rays as a whole have a wide

range of diets; however, each taxon has specific food preferences (see Cortés et al., 2008; Klimley, 2013) that could be used to infer dietary strategies of their fossil relatives (e.g., Carrillo–Briceño et al., 2016a). Information regarding feeding ecology (dietary composition and behavior) of extant/relative species of the taxa recorded in the Cocinetas assemblages (Table S3) was compiled from Cortés et al. (2008); Compagno et al. (2005); Voigt and Weber (2011); Ebert and Stehmann (2013); and the FishBase website (Froese and Pauly, 2017).

Analyses of $\delta^{18}O_{PO4}$ were made in the Stable Isotope Laboratory at University of Lausanne (UNIL) (Table 2). Powder samples of shark teeth enameloid were obtained by abrasion of the crown surface using a micro–drill, and micro fragment samples were obtained by cutting the tip of teeth. In a few cases when only small or fragmented teeth were available bulk samples were taken (enameloid and dentine). Based on previous studies, data still provide valuable information about ecological features of sharks along geochronological sequences (Fischer et al., 2012, 2013a, b; Kocsis et al., 2014; Leuzinger et al., 2015;

Aguilera et al., 2017a). All samples were cleaned in ultrapure water in an ultrasonic bath to reduce sedimentary contamination. International reference of NBS–120c phosphorite and in–house laboratory standards were prepared parallel with the batch. Pretreatment followed the method described by Koch et al. (1997), where powdered teeth were first washed in 1M acetic acid–Ca acetate (pH = 4.5, 2h) to remove any exogenous carbonates, then were thoroughly rinsed several times in ultrapure water. To obtain the $\delta^{18}O_{PO4}$ values the phosphate group in apatite was separated via precipitation as silver phosphate (O'Neil et al., 1994;

Dettman et al., 2001; Kocsis, 2011). The method was adapted from the last review on silver phosphate microprecipitations by



Mine et al. (2017). Triplicates or duplicates of each $Ag_3PO_4$ sample were analyzed on a TC/EA (high–temperature conversion elemental analyzer) (Vennemann et al., 2002) coupled to a Finnigan MAT 253 mass spectrometer, where silver phosphate is converted to CO at 1450 °C via reduction with graphite. Measurements were corrected to in–house $Ag_3PO_4$ phosphate standards (LK–2L: 12.1 ‰ and LK–3L: 17.9 ‰) that had better than ±0.3 ‰ ($1\sigma$) standard deviations during measurements. The

NBS–120c phosphorite reference material had an average value of 21.7 ‰ ±0.1 ‰ ($n = 6$). The isotope ratios are expressed in the $\delta$–notation relative to Vienna Standard Mean Ocean Water (VSMOW).

The $\delta^{18}O_{PO4}$ values in shark teeth is a well known environmental proxy, especially when enameloid derived samples are employed (Vennemann et al., 2001; Zazzo et al., 2004a, b; Lécuyer, 2004; Kocsis, 2011). Longinelli and Nuti (1973a, b) were the first who recognized that the $\delta^{18}O_{PO4}$ values of several ectothermic fishes are related to two environmental parameters: the

water temperature (T) and the $\delta^{18}O$ value of the water ($\delta^{18}O_w$). Based on these studies, an equation that empirically represents the oxygen isotope fractionation between biogenic phosphate and water was suggested ([T (°C) = 111.4 – 4.3 ($\delta^{18}O_{PO4}$ – $\delta^{18}O_w$)]), which was later revised (Kolodny et al., 1983; Pucéat et al., 2010; Lécuyer et al., 2013). This equation is used by paleontologists as a paleothermometer (Barrick et al., 1993; Lécuyer et al., 1993, 1996). Recently the $\delta^{18}O_{PO4}$ values have also been used to estimate the horizontal migrations of fishes into brackish sub–environments (Kocsis et al., 2007; Klug et al.,

2010; Fischer et al., 2012, 2013a, b; Leuzinger et al., 2015).

Paleotemperatures from the $\delta^{18}O_{PO4}$ values were also calculated using the latest equation of Lécuyer et al. (2013) [T (°C) = 117.4 – 4.5 × ($\delta^{18}O_{PO4}$ – $\delta^{18}O_w$)]. For the late Pliocene samples (Ware Formation) a seawater value of 0 ‰ (VSMOW: Vienna Standard Mean Ocean Water), while for the early–middle Miocene samples (Patsúa assemblage, Jimol and Castilletes) a value of –0.4 ‰ was used following estimates of the global seawater isotopic composition (Lear et al., 2000; Billups and

Schrag, 2002).

## 3   Geological and Stratigraphic setting

### 3.1   Jimol Formation (Burdigalian)

This formation is one of the most extensive Cenozoic units in the Cocinetas Basin (Fig. 1b), with a thickness of approximately 203 m, although this composiste section was poorly exposed in the middle parts of the Formation (Moreno et al., 2015). The

lower and upper contacts of the Jimol Formation are conformable with the Uitpa and Castilletes Formations respectively (Fig. 1b). According to Moreno et al. (2015) and Hendy et al. (2015), the unit is characterized by coarse detritic and calcareous lithologies with fewer interbedded muddy levels deposited in a shallow marine paleoenvironment, in inner shelf depth (< 50 m). Abundant invertebrates (Hendy et al., 2015) and some vertebrate remains (Moreno et al., 2015; Moreno–Bernal et al., 2016) have been recorded. A late early Miocene (17.9–16.7 Ma) age is assigned to the unit on the basis of macroinvertebrate

biostratigraphy and $^{87}Sr/^{86}Sr$ isotope chronostratigraphy (see Hendy et al., 2015).



### 3.2 Castilletes Formation (Burdigalian–Langhian)

This geological unit crops out along the eastern margin of the Cocinetas Basin (Fig. 1b). The lithology of the Castilletes Formation is characterized by successions of mudstones interbedded with thin beds of biosparites and sandstones, with an estimated thickness of 440 m, being the lower contact conformable with the underlying Jimol Formation, and the upper is

unconformable (angular contact) with the overlying Ware Formation (Moreno et al., 2015). The unit was deposited in shallow marine to fluvio–deltaic environments, with abundant marine, fluvio–lacustrine and terrestrial fossils (e.g., plants, mollusks, crustaceans, fishes, turtles, crocodilians, and mammals) (Aguilera et al., 2013, 2017b; Cadena and Jaramillo, 2015; Hendy et al., 2015; Moreno et al., 2015; Amson et al., 2016; Moreno–Bernal et al., 2016; Aguirre–Fernández et al., 2017). Isotope chronostratigraphy ($^{87}$Sr/$^{86}$Sr) supports an age of 16.2 Ma (range: 16.33–16.07) for the lower section, and 15.30 Ma (range:

15.14–15.43) for the middle part of the unit (Moreno et al., 2015).

### 3.3 Undifferentiated Jimol and Castilletes Formation (Burdigalian–Langhian)

Sediments of Bahia Cocinetas in the Patsúa Valley have been previously mapped as the Castilletes Formation (Moreno et al., 2015; Moreno–Bernal et al., 2016). They unconformably overly carbonates of the Siamana Formation (late Oligocene–early Miocene), and are in turn overlain with an angular unconformity by the Ware Formation along the shoreline of Bahia Cocinetas.

Despite these stratigraphic relationships, this succession cannot be physically correlated with any particular beds in either Jimol or Castilletes Formations in the central and northern parts of Cocinetas Basin. The lithofacies preserved in this succession, which includes fossiliferous conglomerate and coarse sands, and distinct fossil assemblages (Teredo–bored wood, an oceanic fauna of mollusks and echinoderms, and diverse elasmobranch and bony fish faunas), are also anomalous. For the purposes of analyzing the biodiversity and paleoecology of elasmobranch faunas in Cocinetas Basin it is best to refer to these beds as

the undifferentiated Jimol/Castilletes Formation. The underlying Siamana Formation may be as young as Aquitanian–early Burdigalian (Silva–Tamayo et al., 2017) thereby constraining the maximum age of these beds as Burdigalian.

### 3.4 Ware Formation (late Pliocene)

The type section of the Ware Formation is located immediately east of the village of Castilletes, and correlated deposits are distributed along the eastern margin of Cocinetas Basin (Fig. 1b), cropping out as conspicuous isolated hills with near hori-

zontal strata (Hendy et al., 2015; Moreno et al., 2015). The lithology of the Ware Formation is composed of light gray mudstones, grayish–yellow fine sandstones, and muddy sandstones, reddish–gray pebbly conglomerates, yellowish–gray packstone biosparites, and sandy to conglomeratic biosparites, with an estimated thickness of approximately 52 m. The lower contact is unconformable with the underlying Castilletes Formation, and the upper contact is a fossiliferous packstone in the stratotype that marks the youngest preserved Neogene sedimentation in the Cocinetas Basin (Moreno et al., 2015; Pérez–Consuegra

et al., 2018). The basal section of the unit was deposited in a fluvio–deltaic environment, and abundant plant and vertebrate remains (including sharks herein referred, fishes, turtles, crocodilians, and mammals) have been found in the conglomeratic layers (Moreno et al., 2015; Amson et al., 2016; Moreno–Bernal et al., 2016; Pérez et al., 2016). Only marine invertebrates



have been found in the top beds of the Ware Formation (e.g., Hendy et al., 2015), suggesting an exposed open–ocean shoreface and nearshore settings near coral reefs (Moreno et al., 2015). A late Pliocene (Piacenzian) range of 3.40 Ma to 2.78 Ma age is assigned to the Ware Formation on the base of macroinvertebrate biostratigraphy and $^{87}$Sr/$^{86}$Sr isotope chronostratigraphy (Moreno et al., 2015).

## 4 Results

### 4.1 Elasmobranch paleodiversity

The taxonomical composition of the 36 fossiliferous localities (Table S1) includes at least 30 taxa of squalomorphs, galeomorphs and batoids (Table 1), Figs. 3–8. Squalomorphs are represented by two species, two genera and two families of Squaliformes and Pristiophoriformes. Galeomorphs are represented by at least 20 species, 13 genera and seven families of Orectolobiformes, Lamniformes and Carcharhiniformes (Table 1). Batoids include seven species, seven genera and seven families of Rhinopristiformes and Myliobatiformes (Table 1).

- **Squaliformes Goodrich, 1909.** This group (Table 1) is represented by two specimens referable to *Dalatias* cf. *D. licha* (Bonnaterre, 1788) (Fig. 3a–d, Table S2) from the Jimol Formation (Table S1). This taxon was previously recorded from the Cocinetas Basin (Uitpa Formation) by Carrillo–Briceño et al. (2016b).

- **Pristiophoriformes Berg, 1958.** Five isolated crowns of rostral teeth of indet. *Pristiophorus* Müller and Henle, 1837 (Fig. 3e–g, Table 1, Table S2), were collected in the Patsúa Valley from the locality 290468 (Table S1). Similar specimens were recorded from the Uitpa Formation by Carrillo–Briceño et al. (2016b).

- **Orectolobiformes Applegate, 1972.** Eight specimens referable to an indet. species of Nebrius Rüppell, 1837 (Fig. 3h–o, Table 1, Table S2), were collected exclusively from Burdigalian localities of the Castilletes Formation (Table S1). The specimens are morphologically similar to those of *Nebrius* sp. reported from the Cantaure Formation (Burdigalian) in the Falcon Basin, Venezuela and Pirabas Formation (Aquitanian–Burdigalian), Brazil (Aguilera et al., 2017a). For summarized information about taxonomy and stratigraphic range of *Nebrius* in the Americas see Carrillo–Briceño et al. (2016a, p. 6).

- **Lamniformes Berg, 1937.** These sharks represent the second most diverse group from the Cocinetas elasmobranch assemblages (Fig. 9a), with records for the Jimol and Castilletes Formations and Patsúa assemblage (locality 290468) (Fig. 9b, Tables S1–S2). *Isurus* cf. *I. oxyrinchus* Rafinesque, 1810 (Fig. 3p–t), †*Paratodus benedenii* (Le Hon, 1871) (Fig. 3u–v), †*Carcharocles chubutensis* (Ameghino, 1901) (Figs. 3w–z, 4a–d), *Alopias* cf. †*A. exigua* (Probst, 1879) (Fig. 4n–q), and †*Anotodus retroflexus* (Agassiz, 1843) (Fig. 4r–s), are recorded exclusively for the locality 290468 (Table S1), whereas *Carcharocles* sp. (Fig. 4m) occurs in the Jimol Formation, and †*Carcharocles megalodon* (Agassiz, 1843) (Fig. 4e–l) from only three localities of the late Burdigalian strata of the Castilletes Formation (Table S1). †*Carcharocles chubutensis* and †*C. megalodon* are the most abundant lamniforms from all studied localities of the Cocinetas Basin (Table S1). Due to the relatively small size of the †*C. chubutensis* teeth from the localities 290468 and 290472, (Table S1), these likely belong to juvenile individuals (Figs. 3w–z, 4a–d).



- **Carcharhiniformes Berg, 1937.** With 14 taxa this is the most diverse and the second most abundant elasmobranch group from the Cocinetas assemblages (Fig. 9a). The Carcharhinidae Jordan and Evermann, 1896 with five genera and 11 species [†*Galeocerdo mayumbensis* Dartevelle and Casier, 1943 (Fig. 4x–z); †*Carcharhinus ackermannii* Santos and Travassos, 1960 (Fig. 5a–d); *Carcharhinus* cf. *C. brachyurus* (Günther, 1870) (Fig. 5e–h); †*Carcharhinus gibbesii* (Woodward, 1889) (Fig. 5k–o); *Carcharhinus leucas* (Müller and Henle, 1839) (Fig. 5p–s); *Carcharhinus* cf. *C. limbatus* (Müller and Henle, 1839) (Fig. 5t–u); *Carcharhinus* cf. *C. perezi* (Poey, 1876) (Fig. 5v–w); *Carcharhinus* cf. †*C. priscus* (Agassiz, 1843) (Figs. 5x–z', 6a–d); †*Isogomphodon acuarius* (Probst, 1879) (Fig. 6h–i); †*Negaprion eurybathrodon* (Blake, 1862) (Fig. 6j–n); †*Physogaleus contortus* (Gibbes, 1849) (Fig. 6o–r)] is the most diverse family represented in the Cocinetas assemblages (Fig. S5). Other less diverse group of carcharhiniforms are represented by Sphyrnidae Gill, 1872 [†*Sphyrna arambourgi* Cappetta, 1970 (Fig. 6s–v); †*Sphyrna laevissima* (Cope, 1867) (Fig. 6w–z')] and Hemigaleidae Hasse, 1879 [†*Hemipristis serra* (Agassiz, 1835) (Fig. 4t–w)], the latter being the most abundant taxon of this group of sharks (Tables S1–S2). From the above referred taxa from the Cocinetas Basin, only †*N. eurybathrodon* shows a record from the early Miocene to the late Pliocene. Although taxonomic discussions are out the scope of this contribution, teeth of †*N. eurybathrodon* are indistinguishable from extant species *Negaprion brevirostris* (Poey, 1868), which also have been noted in the fossil record of the Americas (see Carrillo–Briceño et al., 2015a, table 2; 2016b, table 2). As there is no detailed revision supporting or rejecting the above assumption, just as Carrillo–Briceño et al. (2016a), we use †*N. eurybathrodon* (for fossil specimens) sustained by the principle of priority of the International Code of Zoological Nomenclature. In reference to the *Carcharhinus* spp. teeth (Fig. 6e–g), we have referred all specimens that are broken, eroded and without any diagnostic features for specific identification.

- **Rhinopristiformes Last, Séret and Naylor, 2016.** Two taxa of this group of batoids are represented in the Cocinetas assemblages (Fig. 9, Table 1, Fig. S5). One of them is represented by few isolated and indet. teeth of *Rhynchobatus* Müller and Henle, 1837 (Fig. 7a–i), which are recorded only for the Castilletes Formation (Table S1). Our *Rhynchobatus* sp. specimens resemble those from the Neogene of Venezuela and other locations of Tropical America (Carrillo–Briceño et al., 2016a; Aguilera et al., 2017a), however, we refrain any taxonomic identification at the species level of our specimens, because the range of dental variation in extant species is unknown, and little is known about fossil species from the Americas (Carrillo–Briceño et al., 2016a). The other taxon is represented by a fragment of rostrum and a few rostral denticles of indet. *Pristis* Linck, 1790 (Fig. 7j–m) from the Castilletes and Ware Formations (Table S1). As noted by Carrillo–Briceño et al. (2015b), rostral fragments and denticles are not diagnostic for accurate specific taxonomic determinations.

- **Myliobatiformes Compagno, 1973.** Represented by five taxa [†*Plinthicus stenodon* Cope, 1869 (Fig. 8u–x) and indet. teeth of *Dasyatis* Rafinesque, 1810 (Fig. 7n–u); *Aetobatus* Blainville, 1816 (Fig. 7v–x); *Aetomylaeus* Garman, 1913 (Fig. 8a–j); and *Rhinoptera* Cuvier, 1829 (Fig. 8k–t)], this group of batoids (Table 1) is the most abundant and the third most diverse elasmobranch representatives of the Cocinetas assemblages (Fig. 9, Tables S1–S2, Fig. S5). Teeth assigned to *Aetobatus* sp., †*P. stenodon* and *Dasyatis* sp. are scarce and only found in the Castilletes Formation and Patsúa assemblage (locality 290468) (Table S1). *Aetomylaeus* sp. is reported only in Jimol and Castilletes Formations, and the locality 290468; whereas, *Rhinoptera* sp. has a record in the Cocinetas assemblages from the early Miocene to the late Pliocene, being the most abundant taxon (Tables S1–S2). More than 419 hardly eroded and broken teeth without any diagnostic features for generic determination have




been assigned to Myliobatoidea indet. (Table S1), however, we do not rule out that these teeth could belong to *Aetomylaeus* or *Rhinoptera*.

## 4.2 Dietary preferences

Although extant representatives of the fossil elasmobranchs present in the Cocinetas assemblages exhibit a wide range of diets, four feeding preferences of benthic–pelagic predators and filter feeders can be noted (Table S3). For the Jimol Formation, the most diverse feeder group is that of the piscivorous (Fig. 10), dominated by carcharhiniforms, lamniforms, and a minority of squaliforms representatives (Table S3). The second most diverse is durophagous/cancritrophic group (mollusk, crustacean, coral feeders), which is the most abundant in the Jimol assemblages (Fig. 10) and dominated mainly by myliobatiforms taxa (Table S3). †*Carcharocles* sp. is the only possible eurytrophic/sarcophagous (diverse prey source: fishes, reptiles, birds, mammals, etc.) representatives of this unit. Like the Jimol Formation, the assemblage of the Castilletes Formation also shows a diversity dominated by piscivorous representatives (Fig. 10), and abundance dominated by the durophagous/cancritrophic group (represented in the Castilletes assemblages mainly by myliobatiforms) (Table S3). In the Castilletes assemblage, †*Carcharocles megalodon* and †*Galeocerdo mayumbensis* are the only representatives of the eurytrophic/sarcophagous, and the filter feeders (diet based mainly on planktonic microorganisms) is represented only by the mobulid †*Plinthicus stenodon*, being the less abundant and diverse groups of the Castilletes assemblages (Fig. 10, Table S3). In contrast with the assemblages of the Jimol and Castilletes Formations, the Patsúa assemblage (localities 290468 and 290472) is characterized by a higher diversity and abundance of piscivorous, followed by durophagous/cancritrophic diets (Fig. 10, Table S3). Eurytrophic/sarcophagous and filter feeders also are represented in the localities 290468 and 290472 (Fig. 10, Table S3). In contrast with elasmobranch diversity of the Jimol, Castilletes and Patsúa assemblages, the assemblage from the Ware Formation shows a low diversity and abundance (Fig. 10, Tables S1–S3).

## 4.3 Stable isotope analysis of shark teeth

The $\delta^{18}O_{PO4}$ values of the 73 shark teeth ranged from 15.7 ‰ to 21.7 ‰ (VSMOW, Table 2). Samples were grouped in accordance with their geochronological position in the stratigraphic column (Fig. 11). Layers containing few teeth and/or very close to adjacent levels were averaged for better representation. Variability of the $\delta^{18}O_{PO4}$ values within the same beds is up to 4 ‰ (e.g., the highest is in the Patsúa assemblage, locality 290468), however, a large variation was not exclusively found in levels where many samples were analyzed (e.g., Castilletes, locality 390093).

Several teeth were available from the Patsúa assemblage ($n = 26$) and these were carefully interpreted since the age of the assemblage is unknown. Still, the ecological data from the seven shark species present on both localities (290468, 290472) can be discussed. The average isotope data from the two stratigraphically uncertain Patsúa levels are very similar (*t* test: $t(24) = 0.275$; p > 0.78), hence can be considered as one whole dataset.

Regarding the Castilletes Formation the mean $\delta^{18}O_{PO4}$ values do differ along the sedimentary profile (Fig. 11a). Isotopic values increase towards the middle Miocene (localities 130024, 430202: 20.4 ±1.0 ‰, $n = 5$), but then decrease in the following intervals (locality 390093: 18.7 ±1.3 ‰, $n = 4$). However, importantly when pairwise Student's *t* tests are performed following





stratigraphic orders then no significant differences are observed between the sample batches that are following each other. Still, the top youngest data of the Castilletes Formation gives the lowest average $\delta^{18}O_{PO4}$ value for this lithostratigraphic unit. When Tukey's pairwise comparison is applied to the data of the Castilletes layers, then the top bed is significantly different from the two middle levels of 290438 and 430202–130024.

In the youngest unit of the Ware Formation low $^{18}O/^{16}O$ were measured for the bull shark *C. leucas* specimens (CL.1–CL.12: 17.6 ±1.1 ‰, *n* = 12, Fig. 11a). Interestingly, when the average data of the Ware beds is compared to the youngest bed of the Castilletes Formation they do not show significant differences (*t* test: *t*(16) = 0.748, p > 0.46).

From older Jimol Formation only two teeth were analyzed, but their average is undistinguishable from that of the overall average value of both the Castilletes and Patsúa assemblages. The three larger assemblages of Patsúa, Castilletes and Ware

can be compared on a boxplot (Fig. 11b). The averages of the first two are undistinguishable; however, both are significantly different from that of Ware dataset. There is one outlier from each of the Patsúa and Castilletes fauna, which are teeth of a †*Carcharocles chubutensis* (290468) and a †*Negaprion eurybathrodon* (390093), respectively.

## 5   Discussion

### 5.1   Diversity and biostratigraphy significance

Of the elasmobranch assemblages described here from the Cocinetas Basin (∼30 taxa) at least half of the fauna is characterized by extinct taxa (Table 1). With the exception of *Alopias* cf. †*A. exigua* (Fig. 4n–q, Tables S1–S2), representing the first record of this taxon from Tropical America, the remaining taxa from the Cocinetas assemblages have been found in other Neogene deposits of the Americas (e.g., Kruckow and Thies, 1990, Purdy et al., 2001, Aguilera and Lundberg, 2010, Cappetta, 2012, Carrillo–Briceño et al., 2014, 2015b, 2016a, Landini et al., 2017; and references therein). From the Cocinetas assemblages, 17

shark taxa (*Nebrius* sp., †*P. benedenii*, †*C. chubutensis*, †*C. megalodon*, *Alopias* cf. †*A. exigua*, †*A. retroflexus*, †*G. mayumbensis*, †*C. ackermannii*, *Carcharhinus* cf. *C. brachyurus*, *C. leucas*, *Carcharhinus* cf. *C. limbatus*, *Carcharhinus* cf. *C. perezi*, *Carcharhinus* cf. †*C. priscus*, †*I. acuarius*, †*N. eurybathrodon*, †*P. contortus*, and †*S. arambourgi*) and seven batoids (*Rhynchobatus* sp., *Pristis* sp., *Dasyatis* sp., *Aetobatus* sp., *Aetomylaeus* sp., *Rhinoptera* sp., and †*P. stenodon*) are reported for the first time from Colombian Neogene deposits. The elasmobranch assemblages of the Jimol and Castilletes Formations and the

Patsúa assemblage, share certain faunal similarity with the fauna previously described from the underlying Uitpa Formation (e.g., Carrillo–Briceño et al., 2016b).

The elasmobranch fauna of the Cocinetas assemblages show a clear differentiation in paleodiversity between the geological units (see Fig. S5). The Castilletes Formation and Patsúa assemblage are the most diverse units of the overall assemblages from the Cocinetas Basin (Tables S1–S2, Fig. S5). In contrast, the Jimol and Ware Formations are the least diverse units

(Tables S1–S2, Fig. S5). These paleodiversity differences between the geological units of the Cocinetas Basin, in fact, could be attributable to: 1) less intensive sampling, and especially to the less systematic sieving of all studied localities (see Material and Methods section) and/or 2) different lithologic, taphonomic and preservational conditions, without leaving aside a direct response to the paleoenvironmental and paleoecological conditions (see the below Paleoenvironments of the Cocinetas Basin




subsection). The Castilletes Formation and Patsúa assemblage preserve one of the most diverse elasmobranch faunas known from the early–middle Miocene of the Americas (Fig. S6).

Of biostratigraphic significance to the elasmobranch fauna of the Cocinetas assemblages is the record of †*C. megalodon*, †*G. mayumbensis*, †*C. gibbesii* and †*C. ackermannii*. The presence of †*C. megalodon* in late Burdigalian sediments of the

Castilletes Formation (localities 130024, 290824 and 430202, Fig. 2b), confirms the presence of this species during late early Miocene, an assertion that too has been discussed previously for another American localities by Carrillo–Briceño et al. (2016a, p. 21, and references therein). The age of the above referred localities of the Castilletes Formation, have been estimated by $^{87}$Sr/$^{86}$Sr isotope stratigraphy (Hendy et al., 2015, fig. 16, tab. 6). In the case of †*C. chubutensis*, this species is restricted to the Patsúa assemblage, which suggests that the previous specimens of †*Carcharocles* sp. referred to the Uitpa Formation

by Carrillo–Briceño et al. (2016b, fig. 4.12–13), could belong to the former species. †*Carcharhinus gibbesii* in Jimol Formation, as well in the Patsúa assemblage is also present in the Burdigalian sediments of the Cantaure Formation in Venezuela (Carrillo–Briceño et al., 2016a). These records from the late part of early Miocene are notable as the last appearance of †*C. gibbesii* has been regarded as Aquitanian (Carrillo–Briceño et al., 2016b). †*Carcharhinus ackermannii* is reported here from the Burdigalian sediments of the Castilletes Formation and Patsúa assemblage (Tables S1–S2). However, previously it has been

exclusively reported from the early Miocene Cantaure (Venezuela) and Pirabas (Brazil) Formations (Santos and Travassos, 1960; Carrillo–Briceño et al., 2016a; Aguilera et al., 2017a). Due to the scarce fossil record of this extinct species, it is difficult to propose a determined biostratigraphic and geographical range. The absence of this species in other geological units, younger than early Miocene in the Americas or other regions, could suggest that this species is restricted to the early Miocene.

In reference to †*Galeocerdo mayumbensis*, still little is known about its distribution and chronostratigraphy, which has been

figured in the scientific literature only from a few early Miocene localities of Africa (Dartevelle and Casier, 1943; Andrianavalona et al., 2015; Argyriou et al., 2015) and South America (Carrillo–Briceño et al., 2016a; Aguilera et al., 2017a). Some taxonomical misidentifications also include †*G. mayumbensis* from the early Miocene of Africa (Cook et al., 2010, fig. 3c), Asia (Patnaik et al., 2014, plate. 2.12), Central America (Pimiento et al., 2013, fig. 4b), and South America (Santos and Travassos, 1960, fig. 3; Reis, 2005, fig. 6; Costa et al., 2009, fig. 1e, 2c). There is not a consensus about unpublished †*G. mayumbensis*

teeth (labelled/collections) and their localities from the eastern coast of the US, which questionably have been assigned to a middle to late Miocene and Pliocene age. The absence of †*G. mayumbensis* in locations younger than early Miocene (with the exception of the above record from US), and the tendency of the overall stratigraphical distribution of †*G. mayumbensis*, including the new referred record of the Castilletes Formation and the Patsúa assemblage (Table S1), could suggest that this extinct tiger shark was probably restricted to the early Miocene with a widespread distribution in tropical environments.

## 5.2 Paleoenvironments of the Cocinetas Basin

### 5.2.1 Faunal assemblage evaluation

The Neogene sedimentary sequence of the Cocinetas Basin has been characterized by a transition from a shallow marine to a fluvio–deltaic paleoenvironment (e.g., Moreno et al., 2015; Pérez–Consuegra et al., 2018). The geological and paleontological




evidence (mainly based on mollusk, see Hendy et al., 2015) of Jimol Formation indicate depositional conditions characterized by a shallow marine environment (inner shelf depth < 50 m). The elasmobranch fauna from the Jimol Formation is characterized by a higher diversity of carchariniforms and lamniforms piscivorous species (Figs. 9–10). However, in this assemblage, the durophagous/cancritrophic representatives are the most abundant, which could support habitat and feeding preferences

of this later group, related mainly with the abundance of potential prey in marginal marine and brackish environments (see Hendy et al., 2015). The elasmobranch fauna from the Castilletes Formation is mainly characterized by carcharhiniforms and myliobatiforms, where more than the 80% of the abundance corresponds to species of durophagous/cancritrophic feeding preferences (Figs. 9–10). Extant representatives, as well as fossils of the elasmobranch species of the Castilletes Formation, suggest that these taxa are closely related to marginal marine and brackish environments (see Carrillo–Briceño et al., 2015a,

2015b, 2016a and references therein). Abundant marine and terrestrial fossils such as plants, mollusks, crustaceans, fishes, turtles, crocodilians, and mammals in the Castilletes Formation (Aguilera et al., 2013; Cadena and Jaramillo, 2015; Hendy et al., 2015; Moreno et al., 2015; Amson et al., 2016; Moreno–Bernal et al., 2016; Aguirre–Fernández et al., 2017), suggest a depositional environment associated to a shallow marine to fluvio–deltaic environment, similar to those habitats that characterize the Neogene Urumaco sequence in Western Venezuela (Carrillo–Briceño et al., 2015b). Similar also to the elasmobranch

fauna from the Urumaco sequence (Carrillo–Briceño et al., 2015b), durophagous/cancritrophic taxa with capacity to triturate hard shells (*Aetomylaeus*, *Rhinoptera* and Myliobatoidea indet.) are the most abundant elasmobranch remains in the Castilletes Formation. This could be related to the abundance of their potential benthic prey of mollusks and crustaceans. As well as the presence of †*Carcharocles megalodon* in the brackish paleoenvironments of the Urumaco sequence (Aguilera and de Aguilera, 2004; Carrillo–Briceño et al., 2015b), its presence in marine/fluvio–deltaic environment of the Castilletes Formation, support

possible physiological capabilities that allowed it to withstand the variations in salinity in estuarine and possibly river mouth habitats (see Carrillo–Briceño et al., 2015b, p. 24). The Patsúa assemblage, especially the locality 290468, is characterized by a high diversity and abundance of carchariniforms and lamniforms piscivorous species (Figs. 9–10). The presence of the lamniform *Isurus* cf. *I. oxyrinchus*, the otodontid †*Paratodus benedenii*, the alopids *Alopias* cf. †*A. exigua* and †*Anotodus retroflexus*, and the pristhiophoriform *Pristiophorus* sp., could suggest fully marine environment. It is supported by the associated bony

fishes (Acanthuridae, Labridae, Scaridae, Sparidae, Sphyraenidae, Balistidae and Diodontidae, (see Fig. S7), corals, bryozoans, echinoderms and mollusks, suggesting a subtidal marine environment with limited influence from major freshwater input (see Hendy et al., 2015). The mollusks and echinoderms, in particular, are distinctive from those of the Jimol and Castilletes Formations that have been extensively sampled in central and eastern parts of the Cocinetas Basin. The Patsúa assemblage preserves a diversity of species that covers fully marine sandy bottom and reef habitats (e.g., *Spondylus*), while freshwater and brackish

water species are absent. Other notable fossils include abundant fragments of wood that contain *Teredolites* (traces of Teredo or shipworm), and *Aturia* (nautiloid), which presumably were washed up onto a more exposed coastal setting. An isolated and incomplete Odontoceti tooth also was recorded in the locality 290472 (specimen MUN–STRI–44517).

In contrast with the diverse early–middle Miocene elasmobranch assemblages of the Jimol and Castilletes Formations, and the Patsúa assemblage, the fauna of the late Pliocene Ware Formation is low in diversity and abundance (Fig. 9, Tables S1–S3,

Fig. S5). In the same conglomeratic–fossiliferous layer where the elasmobranch come from, abundant vertebrate fishes, turtles,



crocodilians, and mammals, also have been found (Moreno et al., 2015; Amson et al., 2016; Moreno–Bernal et al., 2016; Pérez et al., 2016). A fluvio–deltaic depositional environment has been described for this basal section of the Ware Formation (Moreno et al., 2015; Pérez–Consuegra et al., 2018). The sharks *Carcharhinus leucas* and †*Negaprion eurybathrodon*, as well the batoids *Pristis* sp. and *Rhinoptera* sp., are the only representative species for this unit (Table S1). All these species are able

to inhabit both marine and brackish environments (see Carrillo–Briceño et al., 2015b, fig. 10). *Carcharhinus leucas* and *Pristis* also have the capacity to enter into rivers and live permanently in freshwater lakes (Voigt and Weber, 2011; Faria et al., 2013).

### 5.3 The shark bioapatite and paleosalinity

Samples with $\delta^{18}O_{PO4}$ values less than 18.4 ‰ are likely to have been formed in waters other than exclusively marine ($\delta^{18}O_w$ = 0 ‰), since the paleotemperatures calculated from much low $\delta^{18}O_{PO4}$ are too high to represent typical shark habitats.

However, fishes which form their bioapatite in a freshwater influenced settings with less than 0 ‰ $\delta^{18}O_w$ values (e. g., rivers, lakes) also have lower $\delta^{18}O_{PO4}$ values at the same temperature of formation (Longinelli and Nuti, 1973a; Kolodny et al., 1983; Kocsis et al., 2007; Fischer et al., 2013a; Leuzinger et al., 2015). Therefore, samples with a such low $\delta^{18}O_{PO4}$ values may indicate the presence of brackish–like environments, due to the mixing of seawater with, for example, river water.

The shark tooth $\delta^{18}O_{PO4}$ values can hence be used to estimate paleoenvironmental and relative salinity conditions for the

Patsúa assemblage and two of the three studied formations: Castilletes and Ware (Fig. 11).

• **Patsúa assemblage.** The samples from the Patsúa assemblage have not been separately dated but the teeth from this locality were in situ and their isotopic composition should represent the sediment deposited somewhere within the Burdigalian and Langhian periods. These shark teeth had predominantly "marine" isotopic compositions with one low $\delta^{18}O_{PO4}$ value measured from a †*Carcharocles chubutensis* specimen (CC.4: 17.4 ±0.3 ‰, Table 2, Fig. 11b). This composition typical for

brackish waters was measured for an extinct species, which has analogous habitat to the recent great white shark (*Carcharodon carcharias*), and most of the isotopic data in the extant and fossil species of this group are characteristic of cold waters, because of its long oceanic migrations and formation of bioapatite in such cold settings (Vennemann et al., 2001; Amiot et al., 2008; Ebert et al., 2013; Aguilera et al., 2017a). Statistical comparisons against the available datasets demonstrate this assemblage as undistinguishable from Castilletes Formation (Fig. 11b). Possibly these paleoenvironments were similar and based on the

$\delta^{18}O_{PO4}$ values, the Patsúa assemblage was deposited mainly under marine conditions. Nevertheless, additional sampling and a precise chronological dating of this assemblage are necessary to improve the paleointerpretation of its isotopic data.

• **Castilletes Fm.** The sedimentary sequence of the Cocinetas Basin is described as a transition from a shallow marine to a fluvio–deltaic paleoenvironment. Like the results from the Patsúa assemblage the $\delta^{18}O_{PO4}$ values are predominantly marine, besides a single tooth of †*Negaprion eurybathrodon* (NG.14: 16.7 ±0.2 ‰, Fig. 11a, b), a species from the same genus of

the modern lemon shark (*Negaprion brevirostris*). Extant individuals of this group inhabit marine inshore areas and commonly migrate through enclosed bays or river mouths, supporting a freshwater influence on the isotopic composition measured. In fact, we expected more samples covering the 'brackish' range, since the fossil assemblage of Castilletes Formation suggests a deltaic influence at this interval (Moreno et al., 2015). Paleobathymetric estimations using mollusk invertebrates have shown that in the Castilletes Formation, the paleoenvironments were alternating quickly along the stratigraphic succession, changing between a





marine setting to a freshwater influenced environment and vice–versa (Hendy et al., 2015). The $\delta^{18}O_{PO4}$ mean values show a minor increase from the base towards the middle section of Castilletes (20.4 ±1.0 ‰, $n = 5$, Fig. 11a), decreasing thereafter to the lowest mean value in this formation (18.7 ±1.3 ‰, $n = 4$). Possibly this indicates regional changes in the paleoenvironment of the shark habitat (e. g., marine to estuarine), but since the overall deviation is overlapping between the localities, more

samples would be required to refine such isotopic fluctuation. Nevertheless, the overall shark isotope data represent those parts of Castilletes Formation when fully marine conditions existed in the region. The few outlier specimens (Fig. 11a, b) clearly indicate the nearby presence of rather brackish conditions into which some sharks ventured. This interpretation is in agreement with the higher resolution mollusks data from the region (Hendy et al., 2015).

• **Ware Fm.** Here the isotope data are significantly different from the result from Patsúa and Castilletes (except vs locality

390093, Fig. 11a, b). The $\delta^{18}O_{PO4}$ values are generally lower in this formation, especially for the bull sharks (*Carcharhinus leucas*, CL.1–CL.12: 17.6 ±1.1 ‰, $n = 12$). This euryhaline species, like the lemon shark, also inhabits in marine inshore zones and occasionally migrates into brackish environments. However, bull sharks are currently well recognized for their ability to persist through coastal sub–environments with brackish conditions, as individuals can also swim hundreds of meters upstream even in freshwater (Ebert et al., 2013). The isotopic range from Ware sharks are in a agreement with the fluvio–deltaic

paleoenvironment of deposition described for this formation (Moreno et al., 2015; Pérez–Consuegra et al., 2018). The two samples of lemon shark relatives have $\delta^{18}O_{PO4}$ values which probably have been formed under distinct marine conditions rather than under fluvial influence (NG.15: 20.7 ±0.1 ‰; NG.16: 20.5 ±0 ‰). The worn appearances of the teeth from the conglomerate beds of the Ware Formation indicate longer transport and hence also probably a mixed, time–averaged fauna originated from different layers of a wider fluvio–deltaic system. Therefore, while the bull shark teeth reflect clear fluvial

conditions, the lemon shark remains may have derived from layers originally deposited in the prodelta or nearby shallow coastal marine beds.

The bull shark teeth are also smaller compared to other specimens (and species) employed in this study. Modern representatives of adult bull sharks normally have anterior teeth around 2 cm in height, a size considerably bigger than our sampled teeth (< 1 cm, Fig. S8). Even taking into consideration more curved and possibly posterior teeth of adult specimens, we estimate

that most of our bull shark $\delta^{18}O_{PO4}$ data were obtained from juvenile and subadult individuals. In previous stable isotope investigations, only samples from young specimens from Lake Nicaragua provided $\delta^{18}O_{PO4}$ values characteristic of a brackish condition (Kocsis et al., 2015; Aguilera et al., 2017a). Nevertheless, our results highlight the ecological importance of the paleoenvironments from Cocinetas Basin for the bull sharks, even suggesting the usage of this coastal zone as a paleonursery habitat. Today, young specimens of this group are known for using brackish lagoons as a nursery ground (e.g., Maracaibo Lake,

Rodríguez, 2001, Tavares and Sánchez, 2012). Moreover, the predominant brackish–like $\delta^{18}O_{PO4}$ values in this species may imply that at least since the late Pliocene they were already adapted to live in waters with reduced salinity and face the constant environmental changes (global and regional) of their paleohabitats.



## 6 Conclusions

• A diverse elasmobranch fauna containing 30 taxa of sharks and rays was identified, with the most diverse groups being respectively Carcharhiniformes and Lamniformes. Fossil assemblage seems to represent the paleoenvironments described for the fossiliferous formations of Cocinetas Basin (Jimol, Castilletes and Ware).

• A distinctive assemblage is reported from undifferentiated facies of the Jimol and Castilletes Formation, and represents a subtidal marine environment with limited freshwater influence.

• The biogenic phosphate $\delta^{18}O_{PO4}$ values of 73 shark teeth were evaluated for the sedimentary sequence of Cocinetas Basin. The isotopic data was used for estimate the paleosalinity (e.g., marine vs brackish vs freshwater) and corroborated the paleoenvironments described for Castilletes and Ware formations.

• A predominant brackish–like $\delta^{18}O_{PO4}$ value was measured for bull sharks, which are probably juveniles, suggesting that at least since the late Pliocene this species was already well adapted to migrate through conditions with reduced salinity.

• More samples and additional proxies are recommended to refine our interpretations. Nevertheless, this multidisciplinary study certainly complements further the knowledge about the paleoenvironmental context and evolution of Tropical America.

*Competing interests.* The authors have declared that no competing interests exist

*Acknowledgements.* This work was supported by Swiss National Science Foundation SNF 31003A–149605 to MRSV and by the Smithsonian Tropical Research Institute (National Geographic Society, Anders Foundation, Gregory D. and Jennifer Walston Johnson, 1923 Fund, Universidad del Norte, and National Science Foundation EAR 0957679 to Carlos Jaramillo). The authors wish to especially thank to Henri Cappetta, Sylvain Adnet, Loic Costeur, Rene Kindlimann, Gustavo Ballen and the Wayuu communities of the Alta Guajira for their generous and important counseling, permission for collection revision and collaboration. Participants of fieldwork in Alta Guajira (2009–2014) are

thanked for their assistance in collection of samples. Special thanks to the Center for Microscopy and Image Analysis of the University of Zurich for their assistance and support performing the scanning electron microscopy analysis. Z. Luz would like to thanks Thiago Nascimento for all technical assistance to build the manuscript file. Last but not least, we are thankful to the Alcaldía Bolivariana de Urumaco, the Universidad Experimental Francisco de Miranda; Mapuka Museum of Universidad del Norte (Barranquilla, Colombia), Natural History Museum of Basel (Switzerland), Paleontological collection of the Institut des Sciences de l' Evolution, University of Montpellier (France)

and Palaeontological Institute and Museum at the University of Zurich for their valuable assistance and for access to comparative material.




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



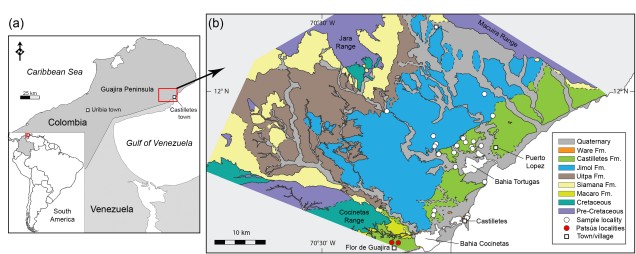

**Figure 1.** Location **(a)** and geological map of the southeastern Cocinetas Basin **(b)**. Abbreviation: Fm. (Formation).





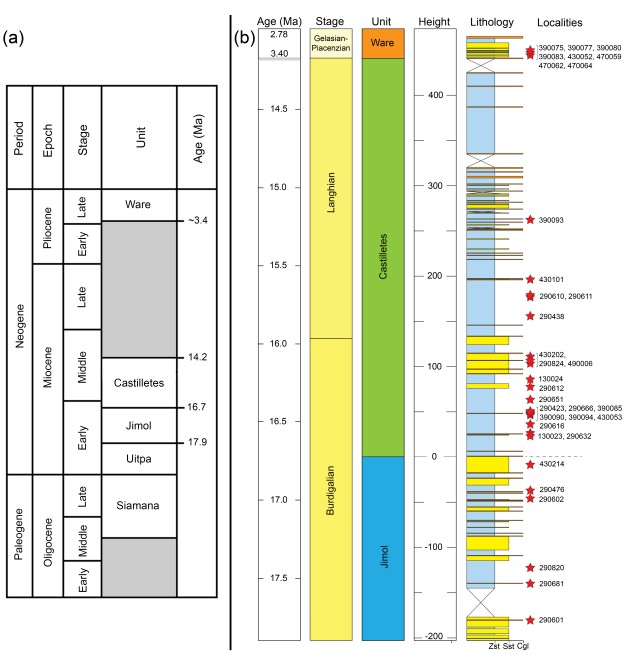

**Figure 2.** Stratigraphy of the Cocinetas Basin. **(a)** Generalized stratigraphy (after Moreno et al., 2015). **(b)** Stratigraphic section and studied localities. Localities of the Patsúa Valley (290468 and 290472) (details in Table S1) are not represented, due these belong to another section of the basin without stratigraphic column.





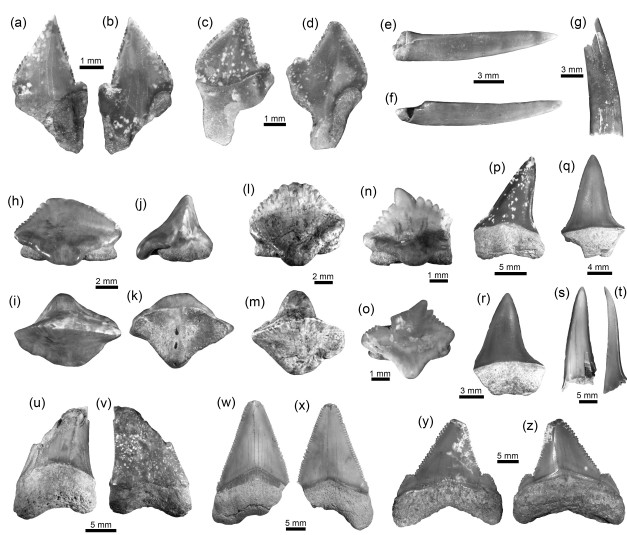

**Figure 3.** Squaliformes, Pristiophoriformes, Orectolobiformes and Lamniformes of the Cocinetas Basin. **(a–d)** *Dalatias* cf. *D. licha* (MUN–STRI–41205). **(e–g)** *Pristiophorus* sp. (MUN–STRI–34788). **(h–o)** Nebrius sp. (h–k, n–o: MUN–STRI–41136; l–m: MUN–STRI–41180). **(p–t)** *Isurus* cf. *I. oxyrinchus* (MUN–STRI–37671). **(u–v)** †*Paratodus benedenii* (MUN–STRI–43742). **(w–z)** †*Carcharocles chubutensis* (MUN–STRI– 40375). Jaw position: upper (y–z?), lower (a–d, w–x) and indet. (h–v), rostral (e–g). View: labial (b, d, h, l, n–o, v, x–y), lingual (a, c, p–s, u, w, z), profile (j, t), occlusal (i, m) dorsal (e–g), and basal (k). Geological unit: Jimol Fm. (a–d), Castilletes Fm. (h–o), Patsúa assemblage–locality 290468 (e–g, p–z).



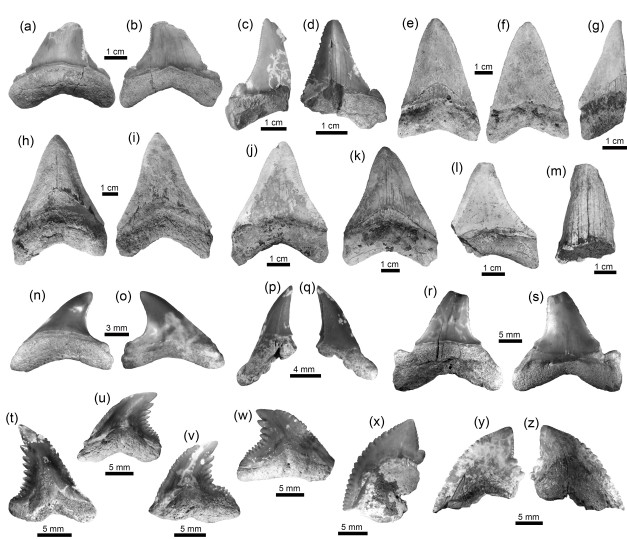

**Figure 4.** Lamniformes and Carcharhiniformes of the Cocinetas Basin. **(a–d)** †*Carcharocles chubutensis* (MUN–STRI–40375). **(e–l)** †*Carcharocles megalodon* (e–g: MUN–STRI–37812; h–i: MUN–STRI–38067; j–l: MUN–STRI–41145). **(m)** †*Carcharocles* sp. (MUN–STRI– 41138). **(n–q)** *Alopias* cf. *A. exigua* (MUN–STRI–43745). **(r–s)** †*Anotodus retroflexus* (MUN–STRI–43740). **(t–w)** †*Hemipristis serra* (MUN–STRI–34790). **(x–z)** †*Galeocerdo mayumbensis* (x: MUN–STRI–41135; y–z: MUN–STRI–40377). Jaw position: upper (j–l, n, u–w), lower (a–b?, c–f, h–i?, p–q?, t) and indet. (g, m, r–s, x–z). View: labial (b–c, f, i–j, l, o, q, s, y), lingual (a, d–e, g–h, k, m–n, p, r, t–x, z). Geological unit: Jimol Fm. (m), Castilletes Fm. (e–l, x), Patsúa assemblage–locality 290468 (a–d, n–w, y–z).





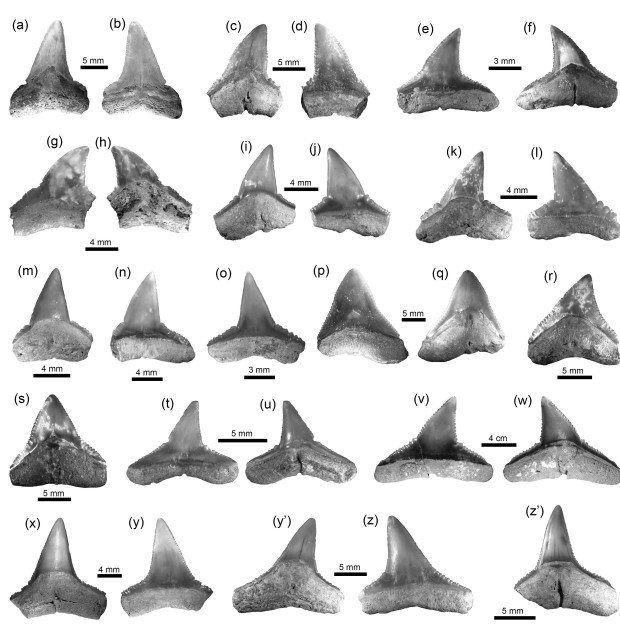

**Figure 5.** Carcharhiniformes of the Cocinetas Basin. **(a–d)** †*Carcharhinus ackermannii* (a–b: MUN–STRI–41128; c–d: MUN–STRI–43743). **(e–h)** *Carcharhinus* cf. *C. brachyurus* (MUN–STRI–41207). **(i–o)** †*Carcharhinus gibbesii* (MUN–STRI–43808). **(p–s)** *Carcharhinus leucas* (p–q: MUN–STRI–37646; r: MUN–STRI–21937; s: MUN–STRI–16287). **(t–u)** *Carcharhinus* cf. *C. limbatus* (MUN–STRI–41153). **(v–w)** *Carcharhinus* cf. *C. perezi* (MUN–STRI–41129). **(x–z')** *Carcharhinus* cf. †*C. priscus* (MUN–STRI–43804). Jaw position: upper (a–z'). View: labial (b, d–e, g, j, l, n–p, t, v, y, z), lingual (a, c, f, h–i, k, m, q–s, u, w–x, y', z'). Geological unit: Jimol Fm. (a–b, e–h, t–w), Castilletes Fm. (t–u). Ware (P–S), Patsúa assemblage–locality 290468 (c–d, i–o, x–z').



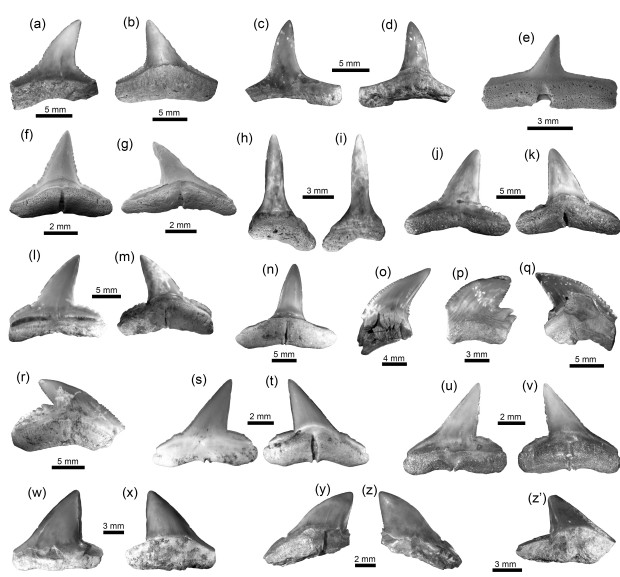

**Figure 6.** Carcharhiniformes of the Cocinetas Basin. **(a–d)** *Carcharhinus* cf. †*C. priscus* (MUN–STRI–43804). **(e–g)** *Carcharhinus* spp. (e: MUN–STRI–42136; f–g: MUN–STRI–42128). **(h–i)** †*Isogomphodon acuarius* (MUN–STRI–41184). **(j–n)** †*Negaprion eury-bathrodon* (MUN–STRI–41133). **(o–r)** †*Physogaleus contortus* (o–q: MUN–STRI–40378; r: MUN–STRI–41132). **(s–v)** †*Sphyrna arambourgi* (MUN–STRI–41143). **(w–z')** †*Sphyrna laevissima* (MUN–STRI–43741). Jaw position: upper (a–b, f–g, j–m, s–z, z'?), lower (c–e, h–i, n) and indet. (o–r). View: labial (a, c, e, i–j, l, p, s, u, w, z), lingual (b, d, f–h, k, m–o, q–r, t, v, x–y, z'). Geological unit: Castilletes Fm. (e–n, r–v), Patsúa assemblage–locality 290468 (a–d, o–q, w–z').





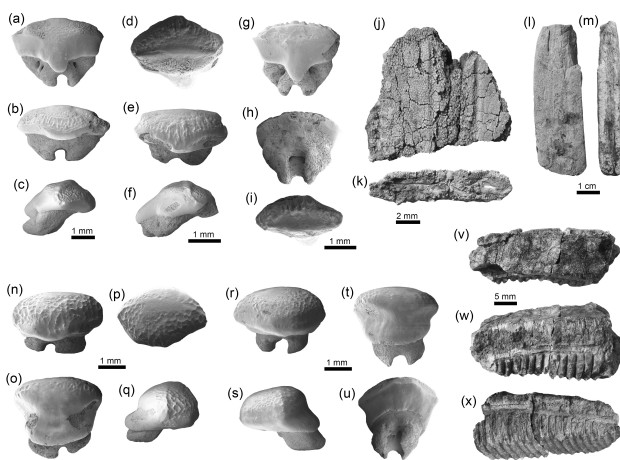

**Figure 7.** Rhinopristiformes and Myliobatiformes of the Cocinetas Basin. **(a–i)** *Rhynchobatus* sp. (MUN–STRI– 42132). **(j–m)** *Pristis* sp. (fragment of rostrum j–k: MUN–STRI–37397; rostral denticle l–m: MUN–STRI–34762). **(n–u)** *Dasyatis* sp. (MUN–STRI–42135). **(v–x)** *Aetobatus* sp. (MUN–STRI–34465). Jaw position: indet. (a–i, n–x). View: labial (b, e, n, r, x), lingual (a, g, o, t, w), profile (c, f, q, s), occlusal (d, i, p, v), dorsal (j, l), posterior (k), basal (h, u). Geological unit: Castilletes Fm. (a–x).





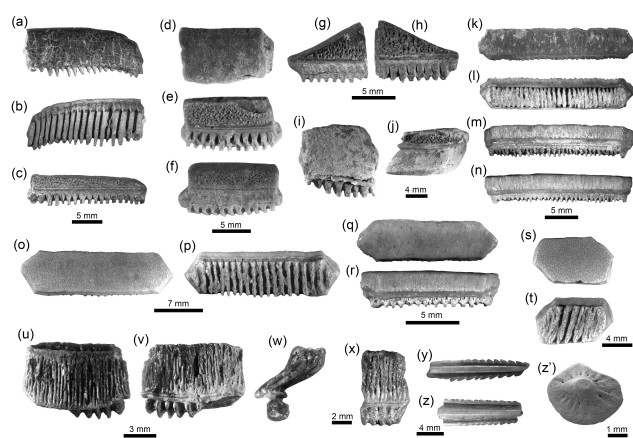

**Figure 8.** Myliobatiformes of the Cocinetas Basin. **(a–j)** *Aetomylaeus* sp. (a–c: MUN–STRI–41134; d–f: MUN–STRI–43746; g–j: MUN–STRI–41134). **(k–t)** *Rhinoptera* sp. (MUN–STRI–41138). **(u–x)** †*Plinthicus stenodon* (MUN–STRI–41203). **(y–z')** Myliobatiformes indet. (caudal spines y–z: MUN–STRI–34785; denticle z': MUN–STRI–42134). Jaw position: indet. (a–x). View: labial (f, g, n, r, u), lingual (c, e, h, m, v, x), profile (j, w), occlusal (a, d, i, k, o, q, s), ventral (y–z), basal (b, l, p, t). Geological unit: Castilletes Fm. (a–c, g–x, z'), Ware Fm. (y–z), Patsúa assemblage–locality 290468 (d–f).



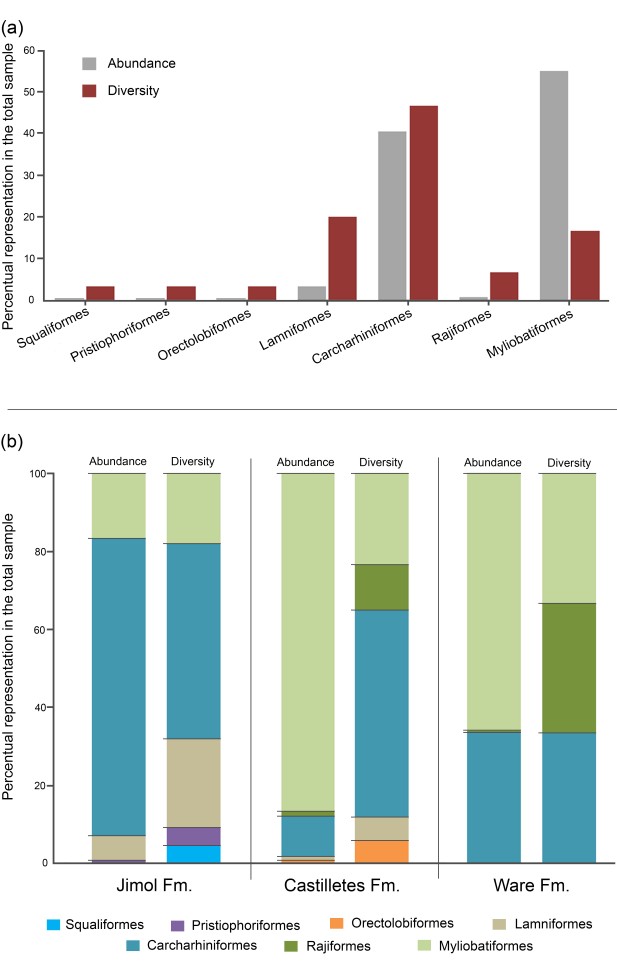

**Figure 9.** Elasmobranch paleodiversity (orders) of the Cocinetas Basin. **(a)** Overall assemblages. **(b)** Assemblages by geological units.




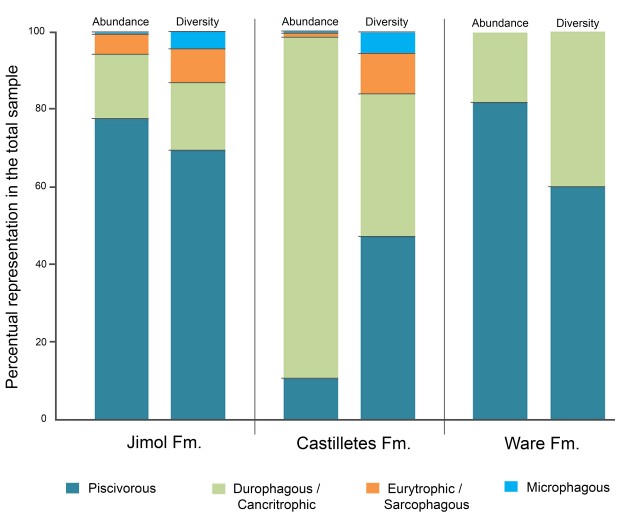

**Figure 10.** Dietary preference of the overall Cocinetas Basin assemblages by geological units.





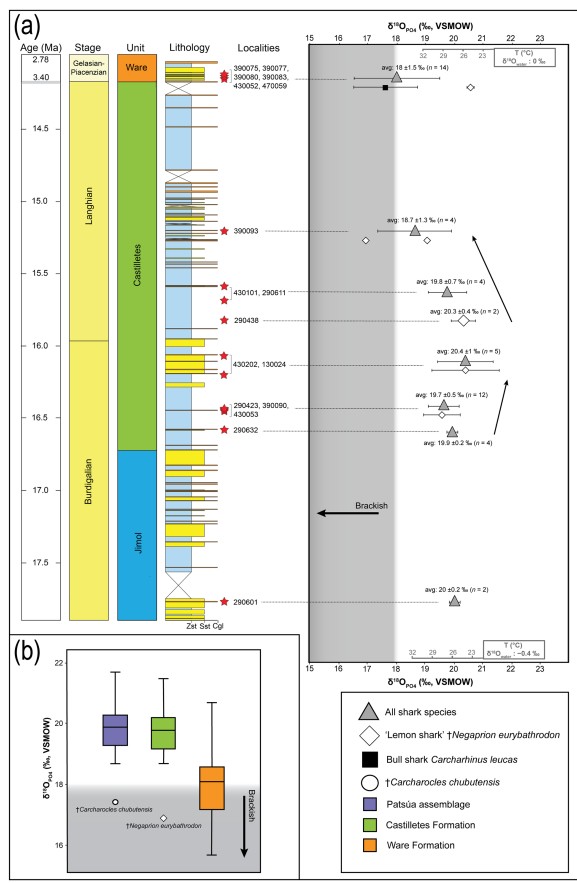

**Figure 11.** Stratigraphic distribution of the $\delta^{18}O_{PO4}$ from sharks of the Cocinetas Basin. The gray–shaded area marks the isotopic range representative of brackish environments. Big symbols give the average of all shark data within the same layer and its standard deviation, while smaller icons are for specific species data. Triangles group all shark species sampled in that layer; while diamonds show the results from †*Negaprion eurybathrodon*, a lemon shark, well represented along the sedimentary sequence (the icon is large for locality 290438 because only *Negaprion* specimens were sampled); and the squares are values from bull sharks of Ware Formation. Temperature bars were estimated from the equation of Lécuyer et al. (2013) are shown at the top (Ware) and at the bottom (Jimol and Castilletes) at $\delta^{18}O_w$ of 0 ‰ and –0.4 ‰, respectively (Lear et al., 2000; Billups and Schrag, 2002). **(a)** The mean $\delta^{18}O_{PO4}$ values show a minor increase along the middle Miocene, with maximum mean value for localities of the late Burdigalian. In the following intervals, the mean values decrease during the early Langhian. Ware Formation samples yielded $\delta^{18}O_{PO4}$ values predominantly characteristic of brackish environments. **(b)** Boxplot of the $\delta^{18}O_{PO4}$ values from samples of the Patsúa assemblage, Castilletes and Ware Formations. Each outlier from the Patsúa assemblage and Castilletes are teeth with $\delta^{18}O_{PO4}$ values considered to form under 'brackish' conditions.





**Table 1.** Elasmobranchii paleodiversity of the Cocinetas Basin.

| Superorder | Order | Family | Genus | Taxon |
|---|---|---|---|---|
| Squalomorphii | Squaliformes | Dalatiidae | *Dalatias* | *Dalatias* cf. *D. licha* (Bonnaterre, 1788) |
| | Pristiophoriformes | Pristiophoridae | *Pristiophorus* | *Pristiophorus* sp. |
| Galeomorphii | Orectolobiformes | Ginglymostomatidae | *Nebrius* | *Nebrius* sp. |
| | Lamniformes | Lamnidae | *Isurus* | *Isurus* cf. *I. oxyrinchus* Rafinesque, 1810 |
| | | †Otodontidae | †*Paratodus* | †*Paratodus benedenii* (Le Hon, 1871) |
| | | | †*Carcharocles* | †*Carcharocles chubutensis* (Ameghino, 1901) |
| | | | | †*Carcharocles megalodon* (Agassiz, 1843) |
| | | | | †*Carcharocles* sp. |
| | | Alopiidae | *Alopias* | *Alopias* cf. *A. exigua* (Probst, 1879) |
| | | | †*Anotodus* | †*Anotodus retroflexus* (Agassiz, 1843) |
| | Carcharhiniformes | Hemigaleidae | *Hemipristis* | †*Hemipristis serra* (Agassiz, 1835) |
| | | Carcharhinidae | *Galeocerdo* | †*Galeocerdo mayumbensis* Dartevelle and Casier, 1943 |
| | | | *Carcharhinus* | †*Carcharhinus ackermannii* Santos and Travassos, 1960 |
| | | | | *Carcharhinus* cf. *C. brachyurus* (Günther, 1870) |
| | | | | †*Carcharhinus gibbesii* (Woodward, 1889) |
| | | | | *Carcharhinus leucas* (Müller and Henle, 1839) |
| | | | | *Carcharhinus* cf. *C. limbatus* (Müller and Henle, 1839) |
| | | | | *Carcharhinus* cf. *C. perezi* (Poey, 1868) |
| | | | | *Carcharhinus* cf. †*C. priscus* (Agassiz, 1843) |
| | | | | *Carcharhinus* spp. |
| | | | †*Isogomphodon* | †*Isogomphodon acuarius* (Probst, 1879) |
| | | | *Negaprion* | †*Negaprion eurybathrodon* (Blake, 1862) |
| | | | †*Physogaleus* | †*Physogaleus contortus* (Gibbes, 1849) |
| | | Sphyrnidae | *Sphyrna* | †*Sphyrna arambourgi* Cappetta, 1970 |
| | | | | †*Sphyrna laevissima* (Cope, 1867) |
| Batomorphii | Rhinopristiformes | Rhynchobatidae | *Rhynchobatus* | *Rhynchobatus* sp. |
| | | Pristidae | *Pristis* | *Pristis* sp. |
| | Myliobatiformes | Dasyatidae | *Dasyatis* | *Dasyatis* sp. |
| | | Aetobatidae | *Aetobatus* | *Aetobatus* sp. |
| | | Myliobatidae | *Aetomylaeus* | *Aetomylaeus* sp. |
| | | Rhinopteridae | *Rhinoptera* | *Rhinoptera* sp. |
| | | | | Myliobatoidea indet. |
| | | Mobulidae | *Plinthicus* | †*Plinthicus stenodon* Cope, 1869 |
| | | | | Myliobatiformes indet. |



**Table 2.** Shark teeth specimens used in geochemical investigation.

| Sample ID | Taxon | Formation | Locality | $\delta^{18}O_{PO4}$ (‰, VSMOW) | $\delta^{18}O_{PO4}$ std dev. |
|---|---|---|---|---|---|
| HS.1 | †*Hemipristis serra* | Jimol | 290601 | 19.9 | 0.1 |
| HS.2 | | | | 20.2 | 0.2 |
| HS.3 | | Patsúa assemblage | 290472 | 20.1 | 0.1 |
| HS.4 | | | | 20 | 0.1 |
| HS.5 | | | | 20.6 | 0.1 |
| CC.1 | †*Carcharocles chubutensis* | | | 19.9 | 0.1 |
| CC.2 | | | | 19.1 | 0.2 |
| CC.3 | | | | 19.4 | 0.1 |
| HS.6 | †*Hemipristis serra* | | 290468 | 19.3 | 0.1 |
| HS.7 | | | | 20.2 | 0.3 |
| HS.8 | | | | 19.9 | 0.1 |
| NG.1 | †*Negaprion eurybathrodon* | | | 18.9 | 0.2 |
| NG.2 | | | | 19.9 | 0.2 |
| GM.1 | †*Galeocerdo mayumbensis* | | | 20.5 | 0.1 |
| GM.2 | | | | 20.3 | 0.1 |
| GM.3 | | | | 19.3 | 0.2 |
| SL.1 | †*Sphyrna laevissima* | | | 19.9 | 0.0 |
| SL.2 | | | | 19.1 | 0.1 |
| SL.3 | | | | 18.7 | 0.3 |
| CC.4 | †*Carcharocles chubutensis* | | | 17.4 | 0.3 |
| CC.5 | | | | 19.2 | 0.2 |
| CC.6 | | | | 20.7 | 0.0 |
| IO.1 | *Isurus* cf. *I. oxyrinchus* | | | 21.7 | 0.3 |
| IO.2 | | | | 20.8 | 0.0 |
| IO.3 | | | | 19.3 | 0.3 |
| PC.1 | †*Physogaleus contortus* | | | 19.8 | 0.0 |
| PC.2 | | | | 20.5 | 0.0 |
| PC.3 | | | | 19.4 | 0.1 |
| HS.9 | †*Hemipristis serra* | Castilletes | 290632 | 19.8 | 0.3 |
| HS.10 | | | | 19.8 | 0.1 |
| CS.1 | *Carcharhinus* sp. | | | 20.1 | 0.2 |
| CS.2 | | | | 20.1 | 0.1 |
| HS.11 | †*Hemipristis serra* | | 290423 | 19.1 | 0.2 |
| NG.3 | †*Negaprion eurybathrodon* | | | 19.5 | 0.3 |
| HS.12 | †*Hemipristis serra* | | 390090 | 19.6 | 0.0 |



**Table 2.** Continued. Shark teeth specimens used in geochemical investigation.

| Sample ID | Taxon | Formation | Locality | $\delta^{18}O_{PO4}$ (‰, VSMOW) | $\delta^{18}O_{PO4}$ std dev. |
|---|---|---|---|---|---|
| HS.13 | †*Hemipristis serra* | Castilletes | 390090 | 19.5 | 0.0 |
| NG.4 | †*Negaprion eurybathrodon* | | | 20.1 | 0.2 |
| NG.5 | | | | 18.8 | 0.2 |
| SA.1 | †*Sphyrna arambourgi* | | | 20.1 | 0.3 |
| SA.2 | | | | 19.2 | 0.1 |
| HS.14 | †*Hemipristis serra* | | 430053 | 20.1 | 0.2 |
| HS.15 | | | | 20.4 | 0.0 |
| NG.6 | †*Negaprion eurybathrodon* | | | 20.4 | 0.1 |
| NG.7 | | | | 19.2 | 0.1 |
| NG.8 | | | 130024 | 19.2 | 0.2 |
| HS.16 | †*Hemipristis serra* | | 430202 | 21.1 | 0.0 |
| HS.17 | | | | 19.7 | 0.1 |
| NG.9 | †*Negaprion eurybathrodon* | | | 21.5 | 0.2 |
| NG.10 | | | | 20.5 | 0.2 |
| NG.11 | | | 290438 | 20.1 | 0.3 |
| NG.12 | | | | 20.6 | 0.1 |
| CS.3 | *Carcharhinus* sp. | | 290611 | 18.9 | 0.2 |
| CS.4 | | | | 20.3 | 0.2 |
| CS.5 | | | | 20.2 | 0.1 |
| HS.18 | †*Hemipristis serra* | | 430101 | 19.8 | 0.1 |
| NG.13 | †*Negaprion eurybathrodon* | | 390093 | 19.1 | 0.1 |
| NG.14 | | | | 16.9 | 0.2 |
| CS.6 | *Carcharhinus* sp. | | | 18.7 | 0.0 |
| CS.7 | | | | 19.9 | 0.1 |
| CL.1 | *Carcharhinus leucas* | Ware | 430059 | 18.1 | 0.1 |
| CL.2 | | | | 18 | 0.1 |
| CL.3 | | | 430052 | 18 | 0.1 |
| CL.4 | | | | 18.4 | 0.0 |
| CL.5 | | | 390083 | 18 | 0.1 |
| CL.6 | | | | 18.9 | 0.0 |
| CL.7 | | | 390080 | 18.6 | 0.1 |
| CL.8 | | | | 15.7 | 0.2 |
| CL.9 | | | 390077 | 15.7 | 0.2 |
| CL.10 | | | | 18.3 | 0.0 |
| CL.11 | | | 390075 | 16.4 | 0.3 |



**Table 2.** Continued. Shark teeth specimens used in geochemical investigation.

| Sample ID | Taxon | Formation | Locality | $\delta^{18}O_{PO4}$ (‰, VSMOW) | $\delta^{18}O_{PO4}$ std dev. |
|---|---|---|---|---|---|
| CL.12 | *Carcharhinus leucas* | Ware | 390075 | 17.2 | 0.2 |
| NG.15 | †*Negaprion eurybathrodon* | | | 20.7 | 0.1 |
| NG.16 | | | | 20.5 | 0.0 |