# Peer review of "Neogene Caribbean elasmobranchs: Diversity, paleoecology and paleoenvironmental significance of the Cocinetas Basin assemblage (Guajira Peninsula, Colombia)"

_Biogeosciences, 2018_

## Author Comment (AC1) · 15 Jun 2018

Dear readers/researchers,

We realized that our figures 9 and 10 were not sent in their last versions. We would like to apologize for the inconvenience and we ask that for the reviewing process, the uploaded figures in this comment are taken into consideration.

Best regards,

[Figure]

Z. Luz

[Figure]

A

[Figure]

B

[Figure]

**Fig. 1.** Figure 09

Percentual representation in the total sample

| | | | |
|---|---|---|---|
| Abundance | Diversity | Abundance | Diversity |

Jimol Fm.    Castilletes Fm.    Ware Fm.    Patsúa assemblage

■ Piscivorous   ■ Durophagous / Cancritrophic   ■ Eurytrophic / Sarcophagous   ■ Microphagous

**Fig. 2.** Figure 10

---

## Referee Comment (RC1) · Anonymous Referee #1 · 9 Jul 2018

Carrillo-Briceño et al. present an interesting chondrichthyan assemblage from northern Colombia with interesting paleontological and geochemical results. The diversity of fauna is impressive and intended/potential scope interesting, but the manuscript in its current form lacks depth to make a compelling research study. I do not go into details in my review because I think these larger and broader issues need to be addressed before an in depth review.

The last paragraph of the introduction is as follows in quotations; I have inserted some thoughts related to these details afterwards.

[Figure]

"A taxonomic revision is presented of the elasmobranch fauna collected in the Cocinetas Basin (Figs. 1–2), from the Jimol (Burdigalian), Castilletes (late Burdigalian–Langhian), Ware (Gelasian–Piacenzian) Formations, and two localities of the Patsúa Valley (Burdigalian–Langhian). " – The authors address this taxanomic revision in <10 lines per family within the results (p6-7) with many families containing more than one taxon. If there are revisions to the taxonomy (or even establishment of taxa or taxon), a more careful description of the specimens, previous taxonomic classification, justification for the changes, and discussion of the systematics are needed at the individual taxa level, either genus or species depending on the classification.

"The assemblage includes 30 taxa, of which 24 are new reports for Colombian Neogene deposits." Again, an assemblage description needs to be more careful and detailed with information on tooth morphology including but not limited to tooth shape, size, position, wear, etc.

"Additionally, paleoecological and paleoenvironmental interpretations based on the feeding ecology of extant counterpart species, as well as estimates of the paleosalinity using stable isotope compositions of oxygen in the bioapatite of shark teeth are discussed." There are no paleosalinity estimates given in this manuscript. There are oxygen isotope values that indicate lower salinity environments, but the authors do not give actual paleosalinity and only refer to broad and qualitative interpretations of environmental conditions. It is possible for the authors to use a paleosalinity model as established in the literature if they use estimates of temperature and freshwater oxygen isotope composition from the literature.

Next, the authors present the generalized diet for modern analogues to discern feeding ecology. However, the authors do not give specific species for modern analogues; many modern families referred to for the fossil specimens have a wide variety of diet and habitat preferences that cannot be easily summarized and condensed as they are in the current manuscript (P. 8 L 4-20). The modern analogues are not identified and furthermore, little to no justification for how and why the fossil taxa should follow

these modern ecological classifications. Further, if the modern analogues were named, I am almost certain that a careful and deeper search of the modern shark ecology research would yield more specifics on dietary preference, migration patterns, and other important aspects of ecology.

"The Cocinetas Basin represents a valuable window into dynamic changes in paleo-diversity experienced by ancient Proto–Caribbean Neogene chondrichthyan faunas." I am sure this chondrichthyan assemblage can yield important information on Proto-Caribbean Neogene chondrichthyan fauna and environmental reconstruction. More detail on the paleontological descriptions and greater context for the stable isotope data are needed to strengthen the paleoecological and paleoenvironmental interpretations.

The authors have a substantial variation in the $\delta 18O$ values from shark teeth. Given the range of Formations, lithology, and likely depositional environments, the results need to be better organized to reflect these differences. In addition, the paleoenvironmental reconstruction based on these oxygen isotope compositions must consider the habitat preference of the shark that is the basis of geochemical analysis. A shark's tooth mineralizes at a fairly fast rate below the epithelium but there is a delay until this tooth reaches the first series within the jaw where it is used and lost (and hence deposited into the fossil record). Therefore, for migratory sharks the $\delta 18O$ value of a tooth may not represent the depositional environment. Parsing out details for modern analogues and their lifestyle can help the authors classify and interpret the variation in $\delta 18O$ values. Two of the co-authors, Kocsis and Venneman, have published widely with stable isotope data from shark teeth. In many of their publications they use modern analogues quite carefully for paleoecological interpretation and paleoenvironmental reconstruction based on geochemical data.

More detailed treatment of paleontological and geochemical data for this chondrichthyan assemblage would strengthen this study. Currently, the goals of this study are not well served due to the qualitative and broad treatment of the data. The

manuscript would also greatly benefit from a stronger "story" that gives more context and framework for its significance related to chondrichthyan paleoecology and evolution; environmental reconstruction; and paleoclimate implications.

---

## Referee Comment (RC2) · DJE Ehret (Referee) · 18 Jul 2018

I have fully reviewed this manuscript and believe that it could be publishable after major revisions. My major concern is with the writing, sentence structure and use of punctuation. I have made substantial comments and suggested revisions in order to refine the manuscript, however there is much work to be done. I would strongly suggest asking Austin Hendy to fully review and comment on the manuscript, as he is a co-author and a colleague. In general, the science behind the manuscript is sound, however the writing does muddle the results, a problem that can be remedied. I think

the manuscript can be shortened somewhat, again by having the language tightened up. With regards to the identifications of the specimens, I mostly agree. I question the Alopias exigua identification, as it has only been recorded from Europe and there are taxa that have previously been reported from the Americas during this time period. I think some discussion (albeit short) on C. megalodon vs. C. chubutensis is needed to inform readers how the authors discern one species from another. Especially in light of their comments regarding FADs for C. megalodon. With regards to the geological interpretations, the use of transgressive/regressive cycles would clarify the authors' interpretations. There are a few places where references to what is known of the habits of C. leucas and N. brevirostris would strengthen the authors' assertations. I would be happy to clarify my comments to the authors or editor and believe with substantial work on the language and writing, this manuscript could be publishable in Bioeosciences.

Please also note the supplement to this comment:
https://www.biogeosciences-discuss.net/bg-2018-271/bg-2018-271-RC2-supplement.pdf

**Supplement:**

[revised manuscript text omitted]

---

## Short Comment (SC1) · 24 Jul 2018

Since I am currently taking part in a cooperative study of the Neogene elasmobranch assemblages of Peru, this preprint drew my attention. Previously published information, original data and observations, and palaeoenvironmental/palaeobiological interpretations are clearly distinct in the manuscript. Systematic attributions of the figured specimens and palaeoecological inferences reflect the modern standards. Interpretations and conclusions are always well-founded and meaningful, and often of large-scale impact. The quality of the figures is top-notch. The bibliography section is exhaustive.

[Figure]

Overall, it is my contention that this paper marks excellence in palaeoichthyological research by investigating a complex assemblage of palaeo-scenarios via the integration of systematic palaeontology and isotope geochemistry, both aspects relying on a solid stratigraphic background. After publication, this work will surely represent a much useful contribution for all those researchers working on the Cainozoic history of elasmobranchs along the coasts of South and Central America.
* * *

---

## Author Comment (AC2) · 16 Aug 2018

**General comments**

Dear Anonymous Referee,

Thank you very much for your considerations about our submitted manuscript. We revised our writing and many sentences were rewritten. We hope that now the manuscript is adequate for publication in the Biogeosciences journal.

Considerations and points raised are answered below:

**Specific comments**

**Comment:** "A taxonomic revision is presented of the elasmobranch fauna collected in the Cocinetas Basin (Figs. 1–2), from the Jimol (Burdigalian), Castilletes (late Burdigalian– Langhian), Ware (Gelasian–Piacenzian) Formations, and two localities of the Patsúa Valley (Burdigalian–Langhian). " – The authors address this taxonomic revision in <10 lines per family within the results (p. 6–7) with many families containing more than one taxon. If there are revisions to the taxonomy (or even establishment of taxa or taxon), a more careful description of the specimens, previous taxonomic classification, justification for the changes, and discussion of the systematics are needed at the individual taxa level, either genus or species depending on the classification.

**Answer:** We are grateful with this important suggestion from the referee. First we want to apologize, because it has been a mistake from us, when we were not clearer in the introduction or methods sections. It generated misunderstandings for the readers. The focus of this manuscript was not a detailed taxonomic revision of the fossil assemblage. For 30 taxa we should dedicate a long description section which could resulted in a long monograph, far for the plan and objectives expected for this manuscript. Any specimens referred in our contribution do not represent a new species or taxon, for which a description is not required. We have linked all the references for the original descriptions of each taxon, other descriptions and their record in Tropical America for supporting our assignations. Usually paleontological and neontological manuscripts with only taxonomic list do not require a detailed description. In our case, we have presented general information for each taxon, detailed and high quality pictures with the best representative specimens for each taxon. Additionally, supplementary information (e. g., Table S2) with information about total number, tooth measurements, jaw position and provenance of all the fossil specimens are provided.

**Comment:** "The assemblage includes 30 taxa, of which 24 are new reports for Colombian Neogene deposits." Again, an assemblage description needs to be more careful and detailed with information on tooth morphology including but not limited to tooth shape, size, position, wear, etc.

**Answer:** Continuing the idea of the above answer, the assemblage from Colombia is not represented by new taxa for the scientific community. It represents new records from the country of taxa that were previously described and referred from other regions of the Caribbean, Tropical America and the Americas in general (see references section). We presented a paleodiversity compilation of the fossil assemblages. Fossil assemblages have different ways to be described, for example: a) with detailed taxonomic description (which is out the focus of our manuscript), b) just as simple taxonomic lists with or without illustrative support, and c) taxonomic lists, with general information about taxonomic comments and information supported by a detailed supplementary and illustrative information. The last one is our case.

**Comment:** There are no paleosalinity estimates given in this manuscript. There are oxygen isotope values that indicate lower salinity environments, but the authors do not give actual paleosalinity and only refer to broad and qualitative interpretations of environmental conditions. It is possible for the authors to use a paleosalinity model as established in the literature if they use estimates of temperature and freshwater oxygen isotope composition from the literature.

**Answer:** Indeed, no net paleosalinity values are given. Since we lack additional proxies for estimating the freshwater oxygen isotope composition (e. g., marine mammal bones), we have chosen to replace the term 'paleosalinity'.

**Changes:** Replaced in P. 1 L. 8; P. 2 L. 21; P. 13, L. 12; P. 15 L. 26.

**Comment:** Next, the authors present the generalized diet for modern analogues to discern feeding ecology. However, the authors do not give specific species for modern analogues; many modern families referred to for the fossil specimens have a wide variety of diet and habitat preferences that cannot be easily summarized and condensed as

they are in the current manuscript (P. 8 L 4-20). The modern analogues are not identified and furthermore, little to no justification for how and why the fossil taxa should follow these modern ecological classifications. Further, if the modern analogues were named, I am almost certain that a careful and deeper search of the modern shark ecology research would yield more specifics on dietary preference, migration patterns, and other important aspects of ecology.

**Answer:** About "However, the authors do not give specific species for modern analogues", one of the most complex topics and challenges in paleoecology is the inferences about paleo diets. How it is referred in the text "Extant sharks and rays exhibit a wide range of diets; however, each taxon has specific food preferences (see Cortés et al., 2008; Klimley, 2013) that could be used to infer dietary strategies of their fossil relatives". A dietary composition and behavior of extant/relative species of the taxa recorded in the Cocinetas assemblages is compiled in the Table S3. Every fossil taxon with living representative is referred with their analogous living species. Fossil taxa without extant representative at species level are compiled according to the preferences of all extant species present in the genus. For extinct species without extant representatives (e. g., families and genera), their paleoecology and potential feeding preferences have been inferred according their fossil record (based on references), tooth morphology and adaptative dental types, diagnostic characters to infer feeding preferences in shark and rays (see Cappetta, 2012, pp. 17-23).

**Comment:** The authors have a substantial variation in the $\delta^{18}$O values from shark teeth. Given the range of Formations, lithology, and likely depositional environments, the results need to be better organized to reflect these differences. In addition, the paleoenvironmental reconstruction based on these oxygen isotope compositions must consider the habitat reference of the shark that is the basis of geochemical analysis. A shark's tooth mineralizes at a fairly fast rate below the epithelium but there is a delay until this tooth reaches the first series within the jaw where it is used and lost (and hence deposited into the fossil record). Therefore, for migratory sharks the $\delta^{18}$O value

of a tooth may not represent the depositional environment.

**Answer:** We have tried to summarize the information about the $\delta^{18}$O of shark's bioapatite formation and incorporation of low $\delta^{18}$O values in the beginning of the discussion about isotopes (P. 13 L. 13–22). A new sentence about the subject was also added in the P. 15 L. 2–3, when referring about *Negaprion* results from Ware Fm.

**Changes:** Sentence added in P. 15 L. 2–3.

**Comment:** Parsing out details for modern analogues and their lifestyle can help the authors classify and interpret the variation in $\delta^{18}$O values.

**Answer:** We have revised some sentences which we mention modern analogues in our stable isotope discussion section. Since *Carcharhinus leucas* is an extant species, only †*Negaprion eurybathrodon* and †*C. chubutensis* needed relevant examples of a modern analogues. For †*C. chubutensis*, *Carcharodon carcharias* is mentioned (P. 13 L. 27–31) and for †*Negaprion eurybathrodon*, *Negaprion brevirostris* is referred (P. 14 L. 6–8).

We hope to have answered all considerations and to have attended the requirements to publish in the Biogeosciences Journal.

Best regards,

Zoneibe Luz.

**Corresponding author, e-mail:** <zoneibe.luz@gmail.com> **Current address:** Université de Lausanne. Institut des dynamiques de la surface terrestre. Lausanne, Vaud, Switzerland

Please also note the supplement to this comment:
https://www.biogeosciences-discuss.net/bg-2018-271/bg-2018-271-AC2-supplement.pdf

[Figure]

**Supplement:**

[revised manuscript text omitted]

**RC 1 (Anonymous Referee)**

Dear Anonymous Referee,

Thank you very much for your considerations about our submitted manuscript. We revised our writing and many sentences were rewritten. We hope that now the manuscript is adequate for publication in the Biogeosciences journal.

5     Considerations and points raised are answered below:

**Interactive comment responses**

**Comment:** "A taxonomic revision is presented of the elasmobranch fauna collected in the Cocinetas Basin (Figs. 1–2), from the Jimol (Burdigalian), Castilletes (late Burdigalian– Langhian), Ware (Gelasian–Piacenzian) Formations, and two localities of the Patsúa Valley (Burdigalian–Langhian). " – The authors address this taxonomic revision in <10 lines per family within

10  the results (p. 6–7) with many families containing more than one taxon. If there are revisions to the taxonomy (or even establishment of taxa or taxon), a more careful description of the specimens, previous taxonomic classification, justification for the changes, and discussion of the systematics are needed at the individual taxa level, either genus or species depending on the classification.

**Answer:** We are grateful with this important suggestion from the referee. First we want to apologize, because it has been

15  a mistake from us, when we were not clearer in the introduction or methods sections. It generated misunderstandings for the readers. The focus of this manuscript was not a detailed taxonomic revision of the fossil assemblage. For 30 taxa we should dedicate a long description section which could resulted in a long monograph, far for the plan and objectives expected for this manuscript. Any specimens referred in our contribution do not represent a new species or taxon, for which a description is not required. We have linked all the references for the original descriptions of each taxon, other descriptions and their record in

20  Tropical America for supporting our assignations. Usually paleontological and neontological manuscripts with only taxonomic list do not require a detailed description. In our case, we have presented general information for each taxon, detailed and high quality pictures with the best representative specimens for each taxon. Additionally, supplementary information (e. g., Table S2) with information about total number, tooth measurements, jaw position and provenance of all the fossil specimens are provided.

25  **Comment:** "The assemblage includes 30 taxa, of which 24 are new reports for Colombian Neogene deposits." Again, an assemblage description needs to be more careful and detailed with information on tooth morphology including but not limited to tooth shape, size, position, wear, etc.

**Answer:** Continuing the idea of the above answer, the assemblage from Colombia is not represented by new taxa for the scientific community. It represents new records from the country of taxa that were previously described and referred from other

30  regions of the Caribbean, Tropical America and the Americas in general (see references section). We presented a paleodiversity compilation of the fossil assemblages. Fossil assemblages have different ways to be described, for example: a) with detailed taxonomic description (which is out the focus of our manuscript), b) just as simple taxonomic lists with or without illustrative support, and c) taxonomic lists, with general information about taxonomic comments and information supported by a detailed supplementary and illustrative information. The last one is our case.

**Comment:** There are no paleosalinity estimates given in this manuscript. There are oxygen isotope values that indicate lower salinity environments, but the authors do not give actual paleosalinity and only refer to broad and qualitative interpretations of environmental conditions. It is possible for the authors to use a paleosalinity model as established in the literature if they use estimates of temperature and freshwater oxygen isotope composition from the literature.

**Answer:** Indeed, no net paleosalinity values are given. Since we lack additional proxies for estimating the freshwater oxygen isotope composition (e. g., marine mammal bones), we have chosen to replace the term 'paleosalinity'.

**Changes:** Replaced in P. 1 L. 8; P. 2 L. 21; P. 13, L. 12; P. 15 L. 26.

**Comment:** Next, the authors present the generalized diet for modern analogues to discern feeding ecology. However, the authors do not give specific species for modern analogues; many modern families referred to for the fossil specimens have a wide variety of diet and habitat preferences that cannot be easily summarized and condensed as they are in the current manuscript (P. 8 L 4-20). The modern analogues are not identified and furthermore, little to no justification for how and why the fossil taxa should follow these modern ecological classifications. Further, if the modern analogues were named, I am almost certain that a careful and deeper search of the modern shark ecology research would yield more specifics on dietary preference, migration patterns, and other important aspects of ecology.

**Answer:** About "However, the authors do not give specific species for modern analogues", one of the most complex topics and challenges in paleoecology is the inferences about paleo diets. How it is referred in the text "Extant sharks and rays exhibit a wide range of diets; however, each taxon has specific food preferences (see Cortés et al., 2008; Klimley, 2013) that could be used to infer dietary strategies of their fossil relatives". A dietary composition and behavior of extant/relative species of the taxa recorded in the Cocinetas assemblages is compiled in the Table S3. Every fossil taxon with living representative is referred with their analogous living species. Fossil taxa without extant representative at species level are compiled according to the preferences of all extant species present in the genus. For extinct species without extant representatives (e. g., families and genera), their paleoecology and potential feeding preferences have been inferred according their fossil record (based on references), tooth morphology and adaptative dental types, diagnostic characters to infer feeding preferences in shark and rays (see Cappetta, 2012, pp. 17-23).

**Comment:** The authors have a substantial variation in the $\delta^{18}O$ values from shark teeth. Given the range of Formations, lithology, and likely depositional environments, the results need to be better organized to reflect these differences. In addition, the paleoenvironmental reconstruction based on these oxygen isotope compositions must consider the habitat reference of the shark that is the basis of geochemical analysis. A shark's tooth mineralizes at a fairly fast rate below the epithelium but there is a delay until this tooth reaches the first series within the jaw where it is used and lost (and hence deposited into the fossil record). Therefore, for migratory sharks the $\delta^{18}O$ value of a tooth may not represent the depositional environment.

**Answer:** We have tried to summarize the information about the $\delta^{18}O$ of shark's bioapatite formation and incorporation of low $\delta^{18}O$ values in the beginning of the discussion about isotopes (P. 13 L. 13–22). A new sentence about the subject was also added in the P. 15 L. 2–3, when referring about *Negaprion* results from Ware Fm.

**Changes:** Sentence added in P. 15 L. 2–3.

**Comment:** Parsing out details for modern analogues and their lifestyle can help the authors classify and interpret the variation in $\delta^{18}$O values.

**Answer:** We have revised some sentences which we mention modern analogues in our stable isotope discussion section. Since *Carcharhinus leucas* is an extant species, only †*Negaprion eurybathrodon* and †*C. chubutensis* needed relevant examples

5 of a modern analogues. For †*C. chubutensis*, *Carcharodon carcharias* is mentioned (P. 13 L. 27–31) and for †*Negaprion eurybathrodon*, *Negaprion brevirostris* is referred (P. 14 L. 6–8).

We hope to have answered all considerations and to have attended the requirements to publish in the Biogeosciences Journal.

Best regards,

Zoneibe Luz.

10 **Corresponding author, e-mail:** <zoneibe.luz@gmail.com> **Current address:** Université de Lausanne. Institut des dynamiques de la surface terrestre. Lausanne, Vaud, Switzerland

---

## Author Comment (AC3) · 16 Aug 2018

**General comments**

Dear Dr. Ehret,

We are much glad and thankful for your considerations to our manuscript. We accepted many suggestions of your submitted complement (Manuscript comments) to improve and adequate our scientific research for publication in the Biogeosciences journal.

[Figure]

Considerations and points raised are answered below:

**Specific comments**

**Comment:** With regards to the identifications of the specimens, I mostly agree. I question the *Alopias exigua* identification, as it has only been recorded from Europe and there are taxa that have previously been reported from the Americas during this time period.

**Answer:** Our assignation is referred as *Alopias* cf. †*A. exigua*, to keep an open nomenclature in base of the doubts with our specimens. We compared our samples from Colombia with fossil and recent specimens of *Alopias* housed in the collection of the University of Montpellier under the supervision and suggestion of Dr. Cappetta. The specimens of †*A. exigua* are very similar to those of Colombia and the geographic record of the taxon during early Miocene is not a barrier: lately other taxa previously describe only from Europe and North America (e.g., †*Isogomphodon acuarius*, †*Sphyrna arambourgi*, †*Carcharhinus gibbesii*), now are known from Tropical America in sediments of the early Miocene.

**Comment:** There are a few places where references to what is known of the habits of *C. leucas* and *N. brevirostris* would strengthen the authors' assertions..

**Answer:** Thanks, we have added few references regarding migration of these sharks into brackish/freshwater.

**Changes:** References added in P. 13 L. 8–9; P. 14 L. 8, L. 28.

**Manuscript comments (Line of referee comment): author's response / author's changes in manuscript** (page and lines in the Revised Manuscript file are indicated)

**Page 1**

**L. 9–10:** Thanks, we wanted to refer to previous studies which suggest the paleoenvironments for the fossiliferous formations of Cocinetas Basin. Sentence was reformulated, still in P. 1 between lines 9–11.

**Page 2**

**L. 29–31:** Sentence rewritten (L. 30–33).

**Page 3**

**L. 25:** Sample quantity added (L. 29).

**L.27:** Yes, bulk samples were used for few layers. The enameloid in these teeth was either worn/fragmented or was available in limited quantity (small teeth, e. g. *Sphyrna*) to reach the quantity of 1 mg for analysis. A single tooth was used per sample, but including some amount of dentine. The quantity of sample was added (L. 31).

**L.28–29:** Kind of data was added and sentence reformulated (L. 31–32).

**L. 30:** It was not distilled water or from the 'UltraPure' brand. We refer to the also called purified/deionized water (resistivity of 18,2 MOhm). Renamed to 'deionized water' (P. 3 L. 34; P. 4 L. 2).

**Page 4**

**L. 19:** 'Used' since we refer to the adopted seawater isotopic values for our paleotemperature calculation. 'Was used' added on L. 21.

**Page 6**

**L. 8:** Parenthesis fixed (L. 11).

**Page 7**

**L. 11:** Thanks, we wanted to refer to 'carcharhiniforms'. Sentence rewritten (L. 13).

**Page 8**

**L. 20:** We wanted to refer to taxa, information added (L. 26).

**L. 25:** Thanks, sentence was poorly written and it was revised. Now between lines 31–32.

**L.27–29:** Sentence rewritten (P. 9 L. 1–2).

**L. 31–33:** Paragraph rewritten (P. 9 L. 7–15).

**Page 9**

**L. 9–10:** Sentence rewritten (L. 20–25).

**Page 10**

**L. 4:** We agree with the reviewer due the biostratigraphic distribution of these species is still polemic. The †*C. chubutensis* and †*C. megalodon* from Colombia have been differentiated mainly by the crown shape and presence/absence of a pair of lateral cusplets. In the †*C. chubutensis* specimens a pair of lateral cusplets that are not separated from the main cusp can be noticed. The shape morphology of the lower teeth of †*C. chubutensis* is narrower than those of the inferior teeth of †*C. megalodon*. The specimens assigned here as †*C. megalodon* have wider crown in lower teeth, are larger and do not have lateral cusplets. New sentence was added (L. 23–27).

**L. 24:** We agree with the reviewer and we have rewritten the sentence. An amended diagnosis for this taxon does not match the objective of our work, but potential and more detailed revision of the worldwide fossil record of †*G. mayumbensis*, could be and interesting topic for futures contributions. Sentence now between P. 11 L. 6–10.

**L. 25:** The authors are not asserting that collections from the eastern coast of the US do not have good locality data. We just refer that there is not a detailed and systematic revision about the †*G. mayumbensis* specimens from eastern coast of the US. This information has not been published yet.

**L. 27:** As we refered above, a detailed information with clear stratigraphic information about †*G. mayumbensis* in the region has not been published yet. The information

given by the reviewer DJE Ehret about the presence of †*G. mayumbensis* at least in the earlier portions of None Valley the section is very important and help to understand about the biocron of this extinct species. Sentence rewritten and now between L. 10–17.

**Page 11**

**L. 3–6:** Sentence was indeed confusing. We wanted to mention to the piscivorous group of carcharhiniforms and lamniforms. Sentence rewritten (P. 11 L. 33, P. 12 L. 1–2).

**L. 8–10:** Sentence was combined with the sentence above (P. 12 L. 6–8).

**L. 17–21:** Sentence about †*C. megalodon* was deleted.

**Page 12**

**L. 11:** Sentence rewritten (P. 13 L. 16).

**L. 21–22:** We wanted to refer to lamniforms, sentence rewritten (P. 13 L. 29–31).

**L. 31:** Sentence rewritten (P. 14 L. 7–8).

**Page 13**

**L. 1:** Sentence rewritten (P. 14 L. 12–13).

**L. 6:** Sentence rewritten (P. 14 L. 17–20).

**L. 24:** Reference added (P. 15 L. 5–6).

**L. 27–28:** Sentence rewritten (P. 15 L. 10–11).

**Page 14**

**L. 3–4:** Same case of our bad sentence in Page 1, L. 9–10. The sentence was rewritten (P. 15 L. 26–27).

**Page 22**

**About the size of Figure 1:** The original file includes the Figure 1 in high resolution.

**Page 23**

In the Figure 2 **(a)**, due the scale of its representation, we prefer use "stages" in a generalized reference (e.g., middle and late). In **(b)**, due the scale of the column and detail of this, we used the name of the "stages".

**About the localities of the Patsúa Valley:** This unit have been mapped and located as it is referred in our Figure 1, but lack of detailed stratigraphic columns and correlations with the other unit known from the area. More details in Moreno et al. (2015).

**Page 24**

**Figure 3:** Switched the teeth images 'l-m' with 'n-o'.

**Page 32**

**Figure 11:** Removed common names (bull shark, lemon shark) from the figure.

We hope to have answered all considerations and to have attended the requirements to publish in the Biogeosciences Journal.

Best regards,

Zoneibe Luz.

**Corresponding author, e-mail:** <zoneibe.luz@gmail.com> **Current address:** Université de Lausanne. Institut des dynamiques de la surface terrestre. Lausanne, Vaud, Switzerland

Please also note the supplement to this comment:
https://www.biogeosciences-discuss.net/bg-2018-271/bg-2018-271-AC3-
supplement.pdf

[Figure]

**Supplement:**

[revised manuscript text omitted]

**RC 2 (Dana Ehret)**

Dear Dr. Ehret,

We are much glad and thankful for your considerations to our manuscript. We accepted many suggestions of your submitted complement (Manuscript comments) to improve and adequate our scientific research for publication in the Biogeosciences

5 journal.

Considerations and points raised are answered below:

**Interactive comment responses**

**Comment:** With regards to the identifications of the specimens, I mostly agree. I question the *Alopias exigua* identification, as it has only been recorded from Europe and there are taxa that have previously been reported from the Americas during this

10 time period.

**Answer:** Our assignation is referred as *Alopias* cf. †*A. exigua*, to keep an open nomenclature in base of the doubts with our specimens. We compared our samples from Colombia with fossil and recent specimens of *Alopias* housed in the collection of the University of Montpellier under the supervision and suggestion of Dr. Cappetta. The specimens of †*A. exigua* are very similar to those of Colombia and the geographic record of the taxon during early Miocene is not a barrier: lately other taxa pre-

15 viously describe only from Europe and North America (e.g., †*Isogomphodon acuarius*, †*Sphyrna arambourgi*, †*Carcharhinus gibbesii*), now are known from Tropical America in sediments of the early Miocene.

**Comment:** There are a few places where references to what is known of the habits of *C. leucas* and *N. brevirostris* would strengthen the authors' assertions..

**Answer:** Thanks, we have added few references regarding migration of these sharks into brackish/freshwater.

20 **Changes:** References added in P. 13 L. 8–9; P. 14 L. 8, L. 28.

**Manuscript comments (Line of referee comment): author's response / author's changes in manuscript** (page and lines in the Revised Manuscript file are indicated)

**Page 1**

**L. 9–10:** Thanks, we wanted to refer to previous studies which suggest the paleoenvironments for the fossiliferous formations

25 of Cocinetas Basin. Sentence was reformulated, still in P. 1 between lines 9–11.

**Page 2**

**L. 29–31:** Sentence rewritten (L. 30–33).

**Page 3**

**L. 25:** Sample quantity added (L. 29).

30 **L.27:** Yes, bulk samples were used for few layers. The enameloid in these teeth was either worn/fragmented or was available in limited quantity (small teeth, e. g. *Sphyrna*) to reach the quantity of 1 mg for analysis. A single tooth was used per sample, but including some amount of dentine. The quantity of sample was added (L. 31).

**L.28–29:** Kind of data was added and sentence reformulated (L. 31–32).

**L. 30:** It was not distilled water or from the 'UltraPure' brand. We refer to the also called purified/deionized water (resistivity

35 of 18,2 MOhm). Renamed to 'deionized water' (P. 3 L. 34; P. 4 L. 2).

**Page 4**

**L. 19:** 'Used' since we refer to the adopted seawater isotopic values for our paleotemperature calculation. 'Was used' added on L. 21.

**Page 6**

**L. 8:** Parenthesis fixed (L. 11).

**Page 7**

**L. 11:** Thanks, we wanted to refer to 'carcharhiniforms'. Sentence rewritten (L. 13).

**Page 8**

**L. 20:** We wanted to refer to taxa, information added (L. 26).

**L. 25:** Thanks, sentence was poorly written and it was revised. Now between lines 31–32.

**L.27–29:** Sentence rewritten (P. 9 L. 1–2).

**L. 31–33:** Paragraph rewritten (P. 9 L. 7–15).

**Page 9**

**L. 9–10:** Sentence rewritten (L. 20–25).

**Page 10**

**L. 4:** We agree with the reviewer due the biostratigraphic distribution of these species is still polemic. The †*C. chubutensis* and †*C. megalodon* from Colombia have been differentiated mainly by the crown shape and presence/absence of a pair of lateral cusplets. In the †*C. chubutensis* specimens a pair of lateral cusplets that are not separated from the main cusp can be noticed. The shape morphology of the lower teeth of †*C. chubutensis* is narrower than those of the inferior teeth of †*C. megalodon*. The specimens assigned here as †*C. megalodon* have wider crown in lower teeth, are larger and do not have lateral cusplets. New sentence was added (L. 23–27).

**L. 24:** We agree with the reviewer and we have rewritten the sentence. An amended diagnosis for this taxon does not match the objective of our work, but potential and more detailed revision of the worldwide fossil record of †*G. mayumbensis*, could be and interesting topic for futures contributions. Sentence now between P. 11 L. 6–10.

**L. 25:** The authors are not asserting that collections from the eastern coast of the US do not have good locality data. We just refer that there is not a detailed and systematic revision about the †*G. mayumbensis* specimens from eastern coast of the US. This information has not been published yet.

**L. 27:** As we refered above, a detailed information with clear stratigraphic information about †*G. mayumbensis* in the region has not been published yet. The information given by the reviewer DJE Ehret about the presence of †*G. mayumbensis* at least in the earlier portions of None Valley the section is very important and help to understand about the biocron of this extinct species. Sentence rewritten and now between L. 10–17.

**Page 11**

**L. 3–6:** Sentence was indeed confusing. We wanted to mention to the piscivorous group of carcharhiniforms and lamniforms. Sentence rewritten (P. 11 L. 33, P. 12 L. 1–2).

**L. 8–10:** Sentence was combined with the sentence above (P. 12 L. 6–8).

**L. 17–21:** Sentence about †*C. megalodon* was deleted.

**Page 12**

**L. 11:** Sentence rewritten (P. 13 L. 16).

**L. 21–22:** We wanted to refer to lamniforms, sentence rewritten (P. 13 L. 29–31).

**L. 31:** Sentence rewritten (P. 14 L. 7–8).

**Page 13**

**L. 1:** Sentence rewritten (P. 14 L. 12–13).

**L. 6:** Sentence rewritten (P. 14 L. 17–20).

**L. 24:** Reference added (P. 15 L. 5–6).

**L. 27–28:** Sentence rewritten (P. 15 L. 10–11).

**Page 14**

**L. 3–4:** Same case of our bad sentence in Page 1, L. 9–10. The sentence was rewritten (P. 15 L. 26–27).

**Page 22**

**About the size of Figure 1:** The original file includes the Figure 1 in high resolution.

**Page 23**

In the Figure 2 **(a)**, due the scale of its representation, we prefer use "stages" in a generalized reference (e.g., middle and late). In **(b)**, due the scale of the column and detail of this, we used the name of the "stages".

**About the localities of the Patsúa Valley:** This unit have been mapped and located as it is referred in our Figure 1, but lack of detailed stratigraphic columns and correlations with the other unit known from the area. More details in Moreno et al. (2015).

**Page 24**

**Figure 3:** Switched the teeth images 'l-m' with 'n-o'.

**Page 32**

**Figure 11:** Removed common names (bull shark, lemon shark) from the figure.

We hope to have answered all considerations and to have attended the requirements to publish in the Biogeosciences Journal.

Best regards,

Zoneibe Luz.

**Corresponding author, e-mail:** <zoneibe.luz@gmail.com> **Current address:** Université de Lausanne. Institut des dynamiques de la surface terrestre. Lausanne, Vaud, Switzerland

---

## Author Comment (AC4) · 16 Aug 2018

**General comments**

Dear Mr. Collareta,

Many thanks for your considerations about our submitted manuscript. We revised our first version and added few references following recommendations. We hope that now the manuscript is more adequate to be a useful contribution about elasmobranchs of South America.

Best regards,

Zoneibe Luz.

**Corresponding author, e-mail:** <zoneibe.luz@gmail.com> **Current address:** Université de Lausanne. Institut des dynamiques de la surface terrestre. Lausanne, Vaud, Switzerland

Please also note the supplement to this comment:
https://www.biogeosciences-discuss.net/bg-2018-271/bg-2018-271-AC4-supplement.pdf

―――――――――――――――

**Supplement:**

[revised manuscript text omitted]

**SC 1 (Alberto Collareta)**

Dear Mr. Collareta,

Many thanks for your considerations about our submitted manuscript. We revised our first version and added few references following recommendations. We hope that now the manuscript is more adequate to be a useful contribution about elasmobranchs of South America.

Best regards,

Zoneibe Luz.

**Corresponding author, e-mail:** <zoneibe.luz@gmail.com> **Current address:** Université de Lausanne. Institut des dynamiques de la surface terrestre. Lausanne, Vaud, Switzerland

---

## Referee Report (RR1)

The revision by Carrillo–Briceño et al. of "Neogene Caribbean elasmobranchs: Diversity, paleoecology and paleoenvironmental significance of the Cocinetas Basin assemblage (Guajira Peninsula, Colombia)" is improved with more careful handling of data interpretations. However, I have concerns with how the term "diversity" is used and what is meant when it (the term "diversity") is used during various passages throughout the text. Furthermore, the authors discuss many significant and important topics (i.e., body size, salinity gradients), but do not go into enough detail with background or context for the reader unfamiliar with these topics to evaluate this assemblage and dataset. This study is quite interdisciplinary and it is likely that others will read this paper who may not have a full understanding of all the concepts, how they fit together, and their implications. Therefore, I think it is important for the authors to treat the data and interpretations carefully and give enough context for the less interdisciplinary reader who may get new ideas or find use from one morsel within this study.

P2 L 14: Please use "taxonomic list" rather than "taxonomic revision" given your response to the earlier comment in my review (where you offer 3 alternatives for describing fossil assemblages in Comment #2).

P8 L4 What is meant by "most diverse feeding group"? (similarly for "shows a diversity" in L8)Does this mean there is the largest range of dietary preferences or there is the greatest number of taxa within this group? This paragraph is confusing in its reference to diverse vs. abundant. Are these two terms interchangeable (i.e., is richness considered as a factor of diversity?) or are they distinct? If diversity is going to be referenced, a paragraph in the introduction laying the framework and significance of diversity, abundance, richness, etc., especially with respect to fossil shark teeth where migration and deposition are important factors is needed.  In addition, some clarity in the methods would also be helpful; how is "diversity" treated/measured when some taxa are identified to species while others are only to the genus level?

P8 L11 I think this is a misuse of the term, "niche." "Eurytrophic/sarcophagous" and "filter feeding" refer to feeding styles or mechanisms whereas ecological "niche" refers to a multidimensional space of environmental factors for a species or population. If the authors want to use "niche" then "feeding niche" would be more appropriate.

P10 L 24 Assertions about small size of teeth related to juvenile individuals and nurseries need to be substantiated. First, the tooth position and size should be reported in the main document rather than the supplementary material for this detail to remain. In addition, the estimated size for the individual can be made based on regressions by Kenshu Shimada with modern species or a white shark/megalodon allometry study in Gottfried et al. 1996 *Great White Sharks* or a Pimiento et al. 2010 in *PLOS One* on white shark nurseries. Finally, the authors should provide some support of size from other sites and discuss the possibility of smaller body size in this taxa/population.

P12 L9-14 (section 5.3) I find this opening paragraph too abrupt to open this section. Perhaps start with a sentence detailing the range of modern oceans, talk about meteoric water having lower values due to Rayleigh distillation, and hence brackish waters have a

gradient that co-varies with salinity. The first sentence has no context for interpretation for the reader without a stable isotope background.

I would also like to see some justification for why the $\delta^{18}O$ value for water was estimated to be 0‰. If these areas are estuarine with freshwater inputs, it is more likely that the environmental water $\delta^{18}O$ value was less than 0‰ and therefore the temperatures indicated in Fig. 11 are inaccurate. Many readers of this paper will not be familiar with these finer details of oxygen isotope composition interpretations so to put temperature estimates where $\delta^{18}O$ values of environmental water are not well justified will be a disservice to future studies looking for temperature in this time and region.

P13 L6-7 "While the overall shark isotope data represent marine conditions during the deposition of the Castilletes Formation…" I think the authors need to be careful in discussing the stable isotope data because they represent the environmental conditions when the enameloid formed, not necessarily when the shark was in the locality (i.e., it takes some time for the tooth to migrate from where it is mineralized in the back of the jaw to its position in the first series and then lost) or the depositional environment since taphonomy needs to be considered (i.e., shark teeth may be re-deposited from other sediments).

---

## Author Response (AR3)

[revised manuscript text omitted]

**RC 1 (Anonymous Referee)**

Dear Anonymous Referee,

Thank you very much for your considerations about our submitted manuscript. We revised our writing and many sentences were rewritten. We hope that now the manuscript is adequate for publication in the Biogeosciences journal.

5 Considerations and points raised are answered below:

**Interactive comment responses**

**Comment:** "A taxonomic revision is presented of the elasmobranch fauna collected in the Cocinetas Basin (Figs. 1–2), from the Jimol (Burdigalian), Castilletes (late Burdigalian– Langhian), Ware (Gelasian–Piacenzian) Formations, and two localities of the Patsúa Valley (Burdigalian–Langhian). " – The authors address this taxonomic revision in <10 lines per family within 10 the results (p. 6–7) with many families containing more than one taxon. If there are revisions to the taxonomy (or even establishment of taxa or taxon), a more careful description of the specimens, previous taxonomic classification, justification for the changes, and discussion of the systematics are needed at the individual taxa level, either genus or species depending on the classification.

**Answer:** We are grateful with this important suggestion from the referee. First we want to apologize, because it has been 15 a mistake from us, when we were not clearer in the introduction or methods sections. It generated misunderstandings for the readers. The focus of this manuscript was not a detailed taxonomic revision of the fossil assemblage. For 30 taxa we should dedicate a long description section which could resulted in a long monograph, far for the plan and objectives expected for this manuscript. Any specimens referred in our contribution do not represent a new species or taxon, for which a description is not required. We have linked all the references for the original descriptions of each taxon, other descriptions and their record in 20 Tropical America for supporting our assignations. Usually paleontological and neontological manuscripts with only taxonomic list do not require a detailed description. In our case, we have presented general information for each taxon, detailed and high quality pictures with the best representative specimens for each taxon. Additionally, supplementary information (e. g., Table S2) with information about total number, tooth measurements, jaw position and provenance of all the fossil specimens are provided.

25 **Comment:** "The assemblage includes 30 taxa, of which 24 are new reports for Colombian Neogene deposits." Again, an assemblage description needs to be more careful and detailed with information on tooth morphology including but not limited to tooth shape, size, position, wear, etc.

**Answer:** Continuing the idea of the above answer, the assemblage from Colombia is not represented by new taxa for the scientific community. It represents new records from the country of taxa that were previously described and referred from other 30 regions of the Caribbean, Tropical America and the Americas in general (see references section). We presented a paleodiversity compilation of the fossil assemblages. Fossil assemblages have different ways to be described, for example: a) with detailed taxonomic description (which is out the focus of our manuscript), b) just as simple taxonomic lists with or without illustrative support, and c) taxonomic lists, with general information about taxonomic comments and information supported by a detailed supplementary and illustrative information. The last one is our case.

**Comment:** There are no paleosalinity estimates given in this manuscript. There are oxygen isotope values that indicate lower salinity environments, but the authors do not give actual paleosalinity and only refer to broad and qualitative interpretations of environmental conditions. It is possible for the authors to use a paleosalinity model as established in the literature if they use estimates of temperature and freshwater oxygen isotope composition from the literature.

**Answer:** Indeed, no net paleosalinity values are given. Since we lack additional proxies for estimating the freshwater oxygen isotope composition (e. g., marine mammal bones), we replaced the term 'paleosalinity'.

**Changes:** Replaced in P. 1 L. 8; P. 2 L. 19; P. 12, L. 11; P. 14 L. 24.

**Comment:** Next, the authors present the generalized diet for modern analogues to discern feeding ecology. However, the authors do not give specific species for modern analogues; many modern families referred to for the fossil specimens have a wide variety of diet and habitat preferences that cannot be easily summarized and condensed as they are in the current manuscript (P. 8 L 4-20). The modern analogues are not identified and furthermore, little to no justification for how and why the fossil taxa should follow these modern ecological classifications. Further, if the modern analogues were named, I am almost certain that a careful and deeper search of the modern shark ecology research would yield more specifics on dietary preference, migration patterns, and other important aspects of ecology.

**Answer:** About "However, the authors do not give specific species for modern analogues", one of the most complex topics and challenges in paleoecology is the inferences about paleo diets. How it is referred in the text "Extant sharks and rays exhibit a wide range of diets; however, each taxon has specific food preferences (see Cortés et al., 2008; Klimley, 2013) that could be used to infer dietary strategies of their fossil relatives". A dietary composition and behavior of extant/relative species of the taxa recorded in the Cocinetas assemblages is compiled in the Table S3. Every fossil taxon with living representative is referred with their analogous living species. Fossil taxa without extant representative at species level are compiled according to the preferences of all extant species present in the genus. For extinct species without extant representatives (e. g., families and genera), their paleoecology and potential feeding preferences have been inferred according their fossil record (based on references), tooth morphology and adaptative dental types, diagnostic characters to infer feeding preferences in shark and rays (see Cappetta, 2012, pp. 17-23).

**Comment:** The authors have a substantial variation in the $\delta^{18}$O values from shark teeth. Given the range of Formations, lithology, and likely depositional environments, the results need to be better organized to reflect these differences. In addition, the paleoenvironmental reconstruction based on these oxygen isotope compositions must consider the habitat reference of the shark that is the basis of geochemical analysis. A shark's tooth mineralizes at a fairly fast rate below the epithelium but there is a delay until this tooth reaches the first series within the jaw where it is used and lost (and hence deposited into the fossil record). Therefore, for migratory sharks the $\delta^{18}$O value of a tooth may not represent the depositional environment.

**Answer:** We have tried to summarize the information about the $\delta^{18}$O of shark's bioapatite formation and incorporation of low $\delta^{18}$O values in the beginning of the discussion about isotopes (P. 12 L. 12–28). A new sentence about the subject was also added in the P. 14 L. 2–4, when referring about *Negaprion* results from Ware Fm.

**Changes:** Sentence added in P. 14 L. 2–4.

**Comment:** Parsing out details for modern analogues and their lifestyle can help the authors classify and interpret the variation in $\delta^{18}O$ values.

**Answer:** We have revised some sentences which we mention modern analogues in our stable isotope discussion section. Since *Carcharhinus leucas* is an extant species, only †*Negaprion eurybathrodon* and †*C. chubutensis* needed relevant examples of a modern analogues. For †*C. chubutensis*, *Carcharodon carcharias* is mentioned (P. 12, L. 32–33) and for †*Negaprion eurybathrodon*, *Negaprion brevirostris* is referred (P. 13 L. 9–11, 26–27).

**RC 2 (Dana Ehret)**

Dear Dr. Ehret,

We are much glad and thankful for your considerations to our manuscript. We accepted many suggestions of your submitted complement (Manuscript comments) to improve and adequate our scientific research for publication in the Biogeosciences journal.

Considerations and points raised are answered below:

**Interactive comment responses**

**Comment:** With regards to the identifications of the specimens, I mostly agree. I question the *Alopias exigua* identification, as it has only been recorded from Europe and there are taxa that have previously been reported from the Americas during this time period.

**Answer:** Our assignation is referred as *Alopias* cf. †*A. exigua*, to keep an open nomenclature in base of the doubts with our specimens. We compared our samples from Colombia with fossil and recent specimens of *Alopias* housed in the collection of the University of Montpellier under the supervision and suggestion of Dr. Cappetta. The specimens of †*A. exigua* are very similar to those of Colombia and the geographic record of the taxon during early Miocene is not a barrier: lately other taxa previously describe only from Europe and North America (e.g., †*Isogomphodon acuarius*, †*Sphyrna arambourgi*, †*Carcharhinus gibbesii*), now are known from Tropical America in sediments of the early Miocene.

**Comment:** There are a few places where references to what is known of the habits of *C. leucas* and *N. brevirostris* would strengthen the authors' assertions..

**Answer:** Thanks, we have added few references regarding migration of these sharks into brackish/freshwater.

**Changes:** References added in P. 12 L. 7–9; P. 13 L. 11, L. 27–29.

**Manuscript comments (Line of referee comment): author's response / author's changes in manuscript** (page and lines in the Revised Manuscript file are indicated)

**Page 1**

**L. 9–10:** Thanks, we wanted to refer to previous studies which suggest the paleoenvironments for the fossiliferous formations of Cocinetas Basin. Sentence was reformulated, still in P. 1 between lines 9–11.

**Page 2**

**L. 29–31:** Sentence rewritten (L. 29–30).

**Page 3**

**L. 25:** Sample quantity added (L. 29).

**L.27:** Yes, bulk samples were used for few layers. The enameloid in these teeth was either worn/fragmented or was available in limited quantity (small teeth, e. g. *Sphyrna*) to reach the quantity of 1 mg for analysis. A single tooth was used per sample, but including some amount of dentine. The quantity of sample was added (L. 31).

**L.28–29:** Kind of data was added and sentence reformulated (L. 31–32).

**L. 30:** It was not distilled water or from the 'UltraPure' brand. We refer to the also called purified/deionized water (resistivity of 18,2 MOhm). Renamed to 'deionized water' (P. 3 L. 33; P. 4 L. 2).

**Page 4**

**L. 19:** 'Used' since we refer to the adopted seawater isotopic values for our paleotemperature calculation. 'Was used' added on L. 21.

**Page 6**

**L. 8:** Parenthesis fixed (L. 11).

**Page 7**

**L. 11:** Thanks, we wanted to refer to 'carcharhiniforms'. Sentence rewritten (L. 13).

**Page 8**

**L. 20:** We wanted to refer to taxa, information added (L. 19).

**L. 25:** Thanks, sentence was poorly written and it was revised. Now between lines 24–26.

**L.27–29:** Sentence rewritten (L. 27–28).

**L. 31–33:** Paragraph rewritten (L. 30–32; P. 9 L. 1–2).

**Page 9**

**L. 9–10:** Sentence rewritten (L. 7–10).

**Page 10**

**L. 4:** We agree with the reviewer due the biostratigraphic distribution of these species is still polemic. The †*C. chubutensis* and †*C. megalodon* from Colombia have been differentiated mainly by the crown shape and presence/absence of a pair of lateral cusplets. In the †*C. chubutensis* specimens a pair of lateral cusplets that are not separated from the main cusp can be noticed. The shape morphology of the lower teeth of †*C. chubutensis* is narrower than those of the inferior teeth of †*C. megalodon*. The specimens assigned here as †*C. megalodon* have wider crown in lower teeth, are larger and do not have lateral cusplets. New sentence was added (L. 10–12).

**L. 24:** We agree with the reviewer and we have rewritten the sentence. An amended diagnosis for this taxon does not match the objective of our work, but potential and more detailed revision of the worldwide fossil record of †*G. mayumbensis*, could be and interesting topic for futures contributions. Sentence now between L. 24–28.

**L. 25:** The authors are not asserting that collections from the eastern coast of the US do not have good locality data. We just refer that there is not a detailed and systematic revision about the †*G. mayumbensis* specimens from eastern coast of the US. This information has not been published yet.

**L. 27:** As we refered above, a detailed information with clear stratigraphic information about †*G. mayumbensis* in the region has not been published yet. The information given by the reviewer DJE Ehret about the presence of †*G. mayumbensis* at least in the earlier portions of None Valley the section is very important and help to understand about the biocron of this extinct species. Sentence rewritten and now between L. 28–34, P. 11 L. 1–2.

**Page 11**

**L. 3–6:** Sentence was indeed confusing. We wanted to mention to the piscivorous group of carcharhiniforms and lamniforms. Sentence rewritten (L. 9–11).

**L. 8–10:** Sentence was combined with the sentence above (L. 14–15).

**L. 17–21:** Sentence about †*C. megalodon* was deleted.

**Page 12**

**L. 11:** Sentence rewritten (L. 12–16).

**L. 21–22:** We wanted to refer to lamniforms, sentence rewritten (L. 33–34; P. 13 L. 1).

**L. 31:** Sentence rewritten (P. 13 L. 9–11).

**Page 13**

**L. 1:** Sentence rewritten (L. 14).

**L. 6:** Sentence rewritten (L. 18–21).

**L. 24:** Reference added (L. 29).

**L. 27–28:** Sentence rerwritten (P. 14 L. 6–8).

**Page 14**

**L. 3–4:** Same case of our bad sentence in Page 1, L. 9–10. The sentence was rewritten (L. 23).

**Page 22**

**About the size of Figure 1:** The original file includes the Figure 1 in high resolution.

**Page 23**

In the Figure 2 **(a)**, due the scale of its representation, we prefer use "stages" in a generalized reference (e.g., middle and late). In **(b)**, due the scale of the column and detail of this, we used the name of the "stages".

**About the localities of the Patsúa Valley:** This unit have been mapped and located as it is referred in our Figure 1, but lack of detailed stratigraphic columns and correlations with the other unit known from the area. More details in Moreno et al. (2015).

**Page 24**

**Figure 3:** Switched the teeth images 'l-m' with 'n-o'.

**Page 32**

**Figure 11:** Removed common names (bull shark, lemon shark) from the figure.

**RC 1 (Anonymous Referee), 2nd comment**

Dear Anonymous Referee,

We are pleased for receiving further considerations to improve our submitted manuscript. We accepted most of the suggestions, some sentences were removed from our text while others were added. We hope that the manuscript is more adequate and proper for publication in the Biogeosciences journal.

Comments are answered below:

**Comment:** P2 L 14: Please use "taxonomic list" rather than "taxonomic revision" given your response to the earlier comment in my review (where you offer 3 alternatives for describing fossil assemblages in Comment 2).

**Answer:** Accepted, thanks for the suggestion.

**Changes:** Replaced in Pag. 2, L. 16.

**Comment:** P8 L4 What is meant by "most diverse feeding group"? (similarly for "shows a diversity" in L8) Does this mean there is the largest range of dietary preferences or there is the greatest number of taxa within this group? This paragraph is confusing in its reference to diverse vs. abundant. Are these two terms interchangeable (i.e., is richness considered as a factor of diversity?) or are they distinct? If diversity is going to be referenced, a paragraph in the introduction laying the framework and significance of diversity, abundance, richness, etc., especially with respect to fossil shark teeth where migration and deposition are important factors is needed. In addition, some clarity in the methods would also be helpful; how is "diversity" treated/measured when some taxa are identified to species while others are only to the genus level?

**Answer:** Thank you for the comment and, indeed, we have not clarified whether the term "diversity" is used as in ecological studies. We have chosen for our submitted manuscript the use of the term "diversity" as the ecological concept of "richness", where the relative abundance between the taxa in the community is not taken in consideration. To properly discuss "diversity" in fossil assemblages as remarked by the reviewer, factors as migration and deposition should be considered. However, our fossil assemblage is not well-represented to perform such estimations, since for some localities few specimens were found/collected. A new sentence was added in the methods section to clarify the reader about this.

**Changes:** Sentence rewritten in Pag. 8, L. 7. Sentence added in Pag. 3, L. 23–27.

**Comment:** P8 L11 I think this is a misuse of the term, "niche." "Eurytrophic/sarcophagous" and "filter feeding" refer to feeding styles or mechanisms whereas ecological "niche" refers to a multidimensional space of environmental factors for a species or population. If the authors want to use "niche" then "feeding niche" would be more appropriate.

**Answer:** Accepted, thanks.

**Changes:** Replaced in Pag. 8, L. 14.

**Comment:** P10 L 24 Assertions about small size of teeth related to juvenile individuals and nurseries need to be substantiated. First, the tooth position and size should be reported in the main document rather than the supplementary material for this detail to remain. In addition, the estimated size for the individual can be made based on regressions by Kenshu Shimada with modern species or a white shark/megalodon allometry study in Gottfried et al. 1996 Great White Sharks or a Pimiento et al. 2010 in PLOS One on white shark nurseries. Finally, the authors should provide some support of size from other sites and discuss the possibility of smaller body size in this taxa/population.

**Answer:** Thanks for the suggestion. Authors will follow the previous recommendation of deleting this assignment for our studied specimens, now considering that a more detailed examination is needed.

**Changes:** Sentences deleted in Pag. 14, L. 7–9, L. 11–12, L. 26. Sentences rewritten in Pag. 14 L. 8, L. 13.

**Comment:** P12 L9-14 (section 5.3) I find this opening paragraph too abrupt to open this section. Perhaps start with a sentence detailing the range of modern oceans, talk about meteoric water having lower values due to Rayleigh distillation, and hence brackish waters have a gradient that co-varies with salinity. The first sentence has no context for interpretation for the reader without a stable isotope background.

**Answer:** Thanks for the suggestion. A sentence was added in the beginning of the section.

**Changes:** Sentence added in Pag. 12, L. 12–16. Minor correction in L. 17–18.

**Comment:** I would also like to see some justification for why the $\delta^{18}$O value for water was estimated to be 0 ‰. If these areas are estuarine with freshwater inputs, it is more likely that the environmental water $\delta^{18}$O value was less than 0 ‰ and therefore the temperatures indicated in Fig. 11 are inaccurate. Many readers of this paper will not be familiar with these finer details of oxygen isotope composition interpretations so to put temperature estimates where $\delta^{18}$O values of environmental water are not well justified will be a disservice to future studies looking for temperature in this time and region.

**Answer:** Thanks for the comment and few sentences were added/rewritten to clear the reader about this. We used a value of 0 ‰ because from the middle Miocene onwards, Antarctic ice-sheets were permanently present at the globe, which increased the global isotopic composition of seawater close to the modern recognized value of 0 ‰ (Lear et al., 2000; Billups and Schrag, 2002; Hoefs, 2015). Generally, for geochemical studies the value adopted is the mean of global seawater at the time period of the studied subject, even when it is located near sources of brackish/freshwater. The precision of this coastal $\delta^{18}$O$_w$ value would require additional proxies such as oxygen isotopes of mammals bones.

**Changes:** Sentence added in Pag. 12, L. 21–27.

**Comment:** P13 L6-7 "While the overall shark isotope data represent marine conditions during the deposition of the Castilletes Formation…" I think the authors need to be careful in discussing the stable isotope data because they represent the environmental conditions when the enameloid formed, not necessarily when the shark was in the locality (i.e., it takes some time for the tooth to migrate from where it is mineralized in the back of the jaw to its position in the first series and then lost) or the depositional environment since taphonomy needs to be considered (i.e., shark teeth may be re-deposited from other sediments).

**Answer:** Thanks, we have rewritten this sentence. We hope also that the lines in the beginning of this section helps the reader to understand why these values were assigned as 'marine' or 'brackish'.

**Changes:** Sentence added in Pag. 13, L. 18–22.

We hope to have answered all comments and considerations and to have attended the requirements of the Biogeosciences journal.

Best regards,

Zoneibe Luz.

**Corresponding author, e-mail:** <zoneibe.luz@gmail.com> **Current address:** Université de Lausanne. Institut des dynamiques de la surface terrestre. Lausanne, Vaud, Switzerland